# The effect of natural infrastructure on water erosion mitigation in the Andes

Veerle Vanacker[1], Armando Molina[1,2], Miluska Rosas-Barturen[1,3], Vivien Bonnesoeur[4,5], Francisco Román-Dañobeytia[4,5], Boris F. Ochoa-Tocachi[5,6,7], Wouter Buytaert[5,6]

[1]Georges Lemaitre Center for Earth and Climate Research, Earth and Life Institute, UCLouvain, Louvain-la-Neuve, Belgium
[2]Programa para el Manejo del Agua y del Suelo (PROMAS), Facultad de Ingeniería Civil, Universidad de Cuenca, Cuenca, Ecuador.
[3]Departamento Académico de Ingeniería, Pontifica Universidad Católica del Perú, Lima, Perú
[4]Consorcio para el Desarrollo de la Ecorregión Andina (CONDESAN), Lima, Perú
[5]Regional Initiative for Hydrological Monitoring of Andean Ecosystems (iMHEA), Lima, Perú
[6]Department of Civil and Environmental Engineering & Grantham Institute – Climate Change and the Environment, London, United Kingdom
[7]ATUK Consultoria Estrategica, Cuenca 01015, Ecuador

*Correspondence to*: Veerle Vanacker (veerle.vanacker@ulouvain.be)

**Abstract.**

To expand the knowledge base on natural infrastructure for erosion mitigation in the Andes, it is needed to move beyond case-by-case empirical studies to comprehensive assessments. This study reviews the state of evidence on the effectiveness of interventions to mitigate soil erosion by water, and is based on Andean case studies published in grey and peer-reviewed literature. Based on a systematic review of 118 case-studies from the Andes, the study addressed the following research questions: (1) Which erosion indicators allow us to assess the effectiveness of natural infrastructure? (2) What is the overall impact of working with natural infrastructure on on-site and off-site erosion mitigation? and (3) Which locations and types of studies are needed to fill critical gaps in knowledge and research?

Three major categories of natural infrastructure were considered: restoration and protection of natural vegetation such as forest or native grasslands, forestation with native or exotic species, and implementation of soil and water conservation measures for erosion mitigation. From the suite of physical, chemical and biological indicators commonly used in soil erosion research, two indicators were particularly relevant: soil organic carbon of topsoil, and soil loss rates at plot scale. The protection and conservation of natural vegetation has the strongest effect on soil quality, with $3.01 \pm 0.893$ times higher soil organic carbon content in the topsoil compared to control sites. Soil quality improvements are significant but lower for forestation and soil and water conservation measures. Soil and water conservation measures reduce soil erosion to $62.1 \pm 9.2$ %, even though erosion mitigation is highest when natural vegetation is maintained. Further research is needed to evaluate whether the reported effectiveness holds during extreme events related to, for example, El Niño–Southern Oscillation.

## 1 Introduction

The Andes Mountains stretch about 8900 km and crosses tropical, subtropical, temperate and arid latitudes. Very few, if any, of the diverse physiographic, climatic and biogeographic regions in the Andes have been preserved from human impact. The area has been inhabited by humans for more than 15000 years (Jantz and Behling, 2012; Keating, 2007). By the mid-20[th] century, all Andean nations with exception of Argentina experienced an exponential population growth that caused substantial migration both within and between national borders (Little, 1981). More than 85 million people lived in the Andean region by 2020, with the Northern Andes being one of the most densely populated mountain regions in the world (Devenish and Gianella, 2012). The demographic growth and a stagnating agricultural productivity per hectare led to an expansion of the total agricultural land area, either upward to steep hillsides at high elevations covered by native grassland-wetlands ecosystems (Velez et al., 2021), or downward to lands east and west of the Andes covered by tropical and subtropical forests (Wunder, 1996). Land abandonment is widespread where smallholders faced unfavorable economic conditions due to restricted land base, limited availability of farm credit and low productivity in fragile agro-ecological environments (Zimmerer, 1993).

The strong latitudinal gradients in climate and vegetation are reflected in the pronounced north-south gradient in natural erosion processes and rates (Latrubesse and Restrepo, 2014; Montgomery et al., 2001). Natural erosion rates are lowest ($< 25$ t km$^{-2}$ yr$^{-1}$) in the hyper-arid and arid regions, but show high temporal variability as a result of extreme events, in particular during warm El Niño–Southern Oscillation (ENSO) conditions or earthquakes (Carretier et al., 2018; Morera et al., 2017). Erosion rates are usually higher (with rates of $> 250$ t km$^{-2}$ yr$^{-1}$) in the humid regions where the catchments are deeply dissected by bedrock river channels, and where landslides are common (Blodgett and Isacks, 2007; Vanacker et al., 2020). Land use and management have significantly altered the magnitude and frequency of erosion events (Restrepo et al., 2015; Tolorza et al., 2014; Vanacker et al., 2007a). Deforestation and agricultural practices (such as soil tillage and cattle grazing) increase erosion rates (Molina et al., 2007; Podwojewski et al., 2002), river sediment loads (Restrepo et al., 2015) and landslide occurrences (Guns and Vanacker, 2014). Changes in smallholders' livelihoods leading to the abandonment of agricultural land have a non-linear impact on soil erosion rates, as they are often associated with an initial increase in soil erosion, followed by a steady decrease in erosion rates on the long term (Harden, 2001).

To tackle soil erosion and mitigate its on-site and off-site effects, governmental and nongovernmental organizations in the Andean countries launched rural development and soil conservation programs in the 1970s and 1980s: for example, the programs by PRONAREG-MAG-ORSTOM and USAID in Ecuador (De Noni et al., 2001), IIDE and USAID in Bolivia (Zimmerer, 1993) and PRONAMACHCS in Peru (Torero Zegarra et al., 2010). The implementation of large-scale soil conservation and management programs and policies required considerable investments in labor and capital (Bilsborrow, 1992; Zimmerer, 1993; Posthumus and De Graaff, 2005). While their direct and indirect environmental benefits have been demonstrated on case-by-case basis (Farley and Bremer, 2017; Romero-Díaz et al., 2019), comprehensive evaluations of

environmental programs rarely reach beyond case-by-case assessments (Bonnesoeur et al., 2019). For example, the PRONAMACHCS program of the Ministry of Agriculture of Peru promoted the implementation of a specific type of

intervention, the infiltration trenches. They consist of dozens of earthen ditches dug over mountain slopes following contour lines with the objective of increasing water infiltration in the soils. They have been implemented in several catchments throughout the country for over three decades, before the impact of these practices was systematically assessed at the regional scale (Vásquez and Tapia, 2011). In a global systematic review, Locatelli et al. (2020) found that case-studies provide evidence that infiltration trenches are effective in reducing surface runoff and laminar erosion at plot scale, but they also highlight that

their impacts on water infiltration are uncertain as well as their effects at catchment scale or on other erosion forms. There is an urgent need to identify which soil conservation and management practices are most effective to combat soil erosion and to mitigate its on-site and off-site effects in the Andean region.

Soil conservation measures are receiving renewed interest in the context of nature-based solutions. They are defined by the

IUCN as "services that nature provides such as peatlands sequestering carbon, lakes storing large water supplies, and floodplains absorbing excess water runoff" (Cohen-Shacham et al., 2016). Natural infrastructure is part of nature-based solutions, and their infrastructure-like function helps to protect, sustainably manage or restore ecosystems while simultaneously providing human well-being and biodiversity benefits. In the Andean context, three large groups of water-related interventions can be identified: interventions based on land use and protective land cover including (1) restoration and

protection of native ecosystems such as montane forests or grasslands and (2) forestation with native or exotic species, and (3) soil and water conservation measures including crop management, conservation tillage, and slow forming terraces, and the implementation of linear elements such as vegetation strips and check dams. Several studies have shown that working with natural infrastructure can help mitigate soil erosion and reduce risks to natural hazards (Vanacker et al., 2014; Cohen-Shacham et al., 2016).

To expand the knowledge base on natural infrastructure for erosion mitigation in the Andes, moving beyond case-by-case empirical studies to comprehensive assessments is needed (Bonnesoeur et al., 2019). This study reviews systematically the state of evidence on the effectiveness of interventions to mitigate soil erosion by water, and is based on Andean case studies published in grey and peer-reviewed literature. This study addresses the following research questions: (1) Which soil erosion

indicators are useful to assess the overall effectiveness of natural infrastructure interventions from empirical studies in the Andes?; (2) What is the overall impact of implementing natural infrastructure on on-site and off-site erosion mitigation?; and, (3) Which locations and types of studies are needed to fill critical gaps in knowledge and research?

## 2 Materials and methods

The systematic review focuses on natural infrastructure interventions that are expected to influence erosion mitigation. We adapted the typology to the Andean region and defined three large groups of interventions: (i) the restoration and protection of native ecosystems, (ii) the forestation with native or exotic species and (iii) the implementation of soil and water conservation measures. We quantified their effect on the mitigation of water erosion by investigating measurable indicators of soil erosion. Besides common indicators of soil erosion such as soil loss rate, sediment yield, water turbidity and runoff coefficient, we also considered measures of soil quality such as soil organic carbon, soil nutrient content and bulk density. The definition of terms, and search criteria are provided in the Supplement A and B, the database structure in Supplement C, and the studies that were included in the systematic review in Supplement D.

Based on the systematic review of published case-studies from the Andean region, we first summarized the current state of knowledge, explored general patterns, and identified research gaps. We applied the reporting guidelines established in PRISMA, the Preferred Reporting Items for Systematic Reviews and Meta-Analyses (Gurevitch et al., 2018; Moher et al., 2015). Then, we performed analyses of variance to explore systematic differences in soil erosion indicators in relation to the interventions in natural infrastructure. Last, we estimated the overall effect of the interventions on soil quality, and on on-site and off-site erosion mitigation.

## 2.1 Literature search

The peer-reviewed literature search was conducted using the Scopus bibliographic database, and targeting studies published between 1980 and 2020. We searched within the article title, abstract and keywords for the following terms:

$$\left[ \begin{array}{c} *erosion \ \text{OR} \ flood* \ \text{OR} \ landslid* \ \text{OR} \ mass \ movement \ \text{OR} \ alluv* \ \text{OR} \ runoff \ \text{OR} \ infiltration \\ \text{OR} \ gully \ \text{OR} \ sediment \ \text{OR} \ deposition \ \text{OR} \ soil \end{array} \right]$$

AND

$$[Andes \ \text{OR} \ Colombia \ \text{OR} \ Venezuela \ \text{OR} \ Ecuador \ \text{OR} \ Peru \ \text{OR} \ Bolivia \ \text{OR} \ Chile \ \text{OR} \ Argentina]$$

AND

$$\left[ \begin{array}{c} *forest* \ \text{OR} \ grazing \ \text{OR} \ grass* \ \text{OR} \ pasture \ \text{OR} \ agriculture \ \text{OR} \ crop \ * \ \text{OR} \ land \ use \ \text{OR} \\ puna \ \text{OR} \ paramo \ \text{OR} \ bofedal* \ \text{OR} \ dam \ \text{OR} \ reservoir \ \text{OR} \ conservation \ \text{OR} \ management \\ \text{OR} \ till* \ \text{OR} \ terraces \ \text{OR} \ irrigation \ \text{OR} \ lake* \ \text{OR} \ hydraulic* \ \text{OR} \\ ancient \ knowledge \ \text{OR} \ archaeology \ \text{OR} \ human \ \text{OR} \ people \ \text{OR} \ anthropogenic \end{array} \right]$$

For the grey literature, we searched in 35 different databases from specialist organizations, public institutions, and local repositories of private and public universities in the Andean region. The above-mentioned search criteria were adapted for the grey literature given the limited search capabilities of some of the databases. Full details on the literature search are provided in the Supplement B, including the complete search terms, the number of records generated for specific searches, and the name,

location, search dates in Scopus and the national and regional databases of research institutions, universities and specialist

organizations. For international peer-reviewed literature, we used a test library of 20 references (Supplement E) that confirmed that the search strings captured relevant literature.

## 2.2 Inclusion and exclusion criteria

The number of studies that were identified, screened, selected and included in the analysis is shown in a PRISMA flow diagram (Fig. 1). Between 10 January and 27 February 2020, we identified 1798 potentially relevant studies: 91 % corresponding to

peer-reviewed articles and 9 % to grey literature. After removing duplicate studies, the dataset was reduced to 813 studies. These records were screened, and articles that fulfilled the following criteria were included in the database: (1) they present quantitative data on soil erosion or soil quality comparing sites with different land use and protective land cover, soil and water conservation measures, or elements of hydraulic regulation, (2) they are experimental studies including observational datasets or are modelling studies that are fully validated with field experiments or measurements, and (3) they were realized in the

Andean region. During the screening stage, we excluded 623 studies because of absence of quantitative on-site or off-site soil erosion or soil quality measurements.

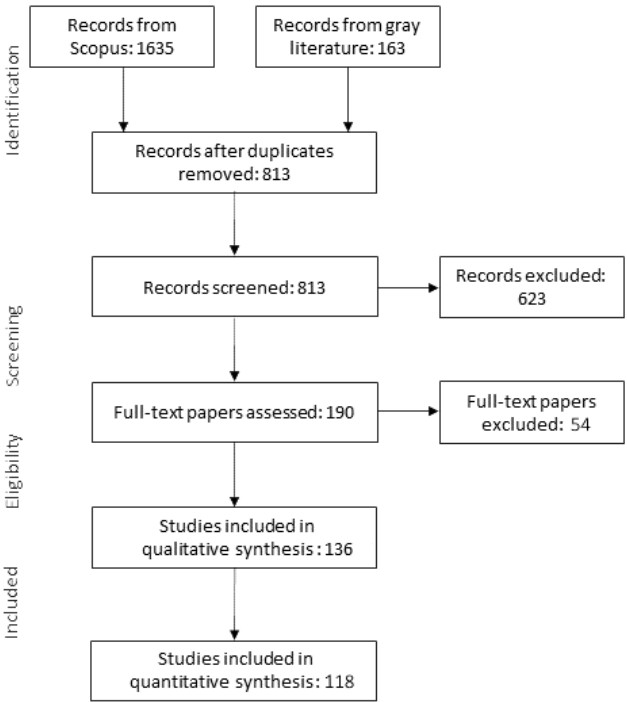

**Figure 1.** Flowchart resuming the results of literature search based on the PRISMA approach

We assessed 190 studies in full-text, and further excluded 54 papers as the studies did not report quantitative measures of erosion rates or soil quality for different classes of land use and protective land cover, soil and water conservation measures, or elements of hydraulic regulation. At this stage, this mainly concerned scientific reports on landslides and landslide-related erosion events.

## 2.3 Database development

A total of 136 studies were included in the systematic review. Where a study encompassed several independent case studies, the case studies were included in the final database as separate entries. Each case study was coded by a unique study identifier and recorded in the georeferenced database (Supplement C). We recorded the following ancillary geographic data: (1) country, (2) site name, (3) coordinates (latitude and longitude in decimal degrees), (4) elevation (meters above sea level, m a.s.l.), and information on (5) bioclimate, (6) surface lithology, (7) ecosystem and (8) landform. The latter four variables were derived
from the 2005 Nature Conservancy datasets via the USGS dataviewer for South America (https://rmgsc.cr.usgs.gov/). We included additional information on the type of study: (8) the experimental design following the classification scheme of Nichols et al. (2011), (9) the modelling approach based on a classification in statistical, process-based and mixed models, (10) the existence of field-data, and (11) the spatial scale and organisation of study based on a classification in plot ($< 0.01$ km$^2$), small catchment (between 0.01 and 1000 km$^2$), large catchment ($> 1000$ km$^2$) and landscape scale (regional) analyses. The latter
contain data collections that are not organised by hydrological units, and that include measurements taken over a larger geographical area.

In the analyses, we quantified the effect of restoration and protection of natural vegetation such as forest or native grasslands (PRO), forestation with native or exotic species (FOR), and implementation of soil and water conservation measures (SWC)
for soil erosion and mitigation (Fig. 2). Soil and water conservation measures (SWC) include crop management, conservation tillage, and slow forming terraces, and the implementation of linear elements such as vegetation strips and check dams. We compared the three natural infrastructure interventions (PRO, FOR and SWC) with untreated areas under traditional agriculture, either cropland (CROP) or rangeland (RANGE), and bare land (BARE). Bare land corresponds to abandoned cropland or degraded land with very low ($< 10$ %) vegetation cover.


The erosion indicators included in this study were (Fig. 2): Sloss = soil loss rate (determined as soil loss in t km$^{-2}$ yr$^{-1}$), RC = plot runoff coefficient (determined as event-based runoff coefficient from rainfall simulation experiments, in %), SSY = specific sediment yield (determined as the catchment-wide sediment yield per surface area, in abandoned cropland or degraded land with very low ($< 10$ %) vegetation cover.  ), and RCC = catchment-wide runoff ratio (determined as the annual total runoff
ratio of the catchment, in %). While Sloss and SSY are direct measures of soil erosion at the plot and catchment scale, the plot and catchment-wide runoff coefficients (RC and RCC) are indirect indicators of soil erosion by water: the rainfall regime plays a role as raindrop impact and runoff water are involved in the detachment of soil particles and transport of sediment in surface

water flow. Empirical studies compiled by, for example, (Bonnesoeur et al., 2019; Valentin et al., 2008) have shown the strong association between runoff coefficients and soil erosion rates.


In addition to the four erosion indicators, two soil quality indicators were included: SOC (total soil organic carbon of the uppermost soil horizon, between 5 and 30 cm, in %), and BD (dry bulk density of the topsoil horizon, between 5 and 30 cm, in g.cm$^{-3}$). Soil organic carbon is the main indicator of soil quality (Franzluebbers, 2002) and directly linked to key soil functions (Wiesmeier et al., 2019) including soil water retention, erosion prevention and resilience to droughts and floods

(Paustian et al., 2016). Bulk density is a commonly reported soil physical property that is related to soil aeration, water and air permeability, and soil microporosity (Horn et al., 1995). Increased bulk density can be indicative of soil compaction, and affect the water retention capacity and accelerate soil erosion (Molina et al., 2007; Patiño et al., 2021). Other erosion indicators were recorded in the database, but not included in the statistical analyses because of a lack of statistical representation. These include plot-based indicators like the stock in SOC over the entire soil depth or the saturated hydraulic conductivity of the topsoil, or

catchment-wide indicators like the presence/relative occurrence of erosion signs or the suspended sediment concentration in the river channels. Mean, sample size and deviation metrics were extracted from figures using PlotDigitizer. Information from in-text tables and supplementary material was copied and tabulated in spreadsheets.

Of the 136 studies included in the systematic review, 118 studies contained sufficient information on the soil erosion and soil

quality indicators to be statistically analyzed. Besides the above-mentioned information, the georeferenced database includes bibliographic details and a URL link to the individual case studies (Supplement D).

### 2.4 Statistical analyses

First, we tested whether sites with natural infrastructure interventions (PRO, FOR and SWC) are different in on-site (Sloss, RC) and off-site (SSY, RCC) soil erosion and soil quality (SOC, BD) compared to untreated areas under traditional agriculture

(CROP, RANGE) or bare land (BARE) as illustrated in Figure 2. The comparison of the 4 erosion and 2 soil quality indicators between the treatments was performed using one-way analysis of variance (ANOVA). In this analysis, we pooled all observations from the 118 case-studies. Because of the limited number of quantitative case studies for the Andes, the number of observations is not the same for each group. Given the low number of observations per group, the Kruskal-Wallis ANOVA on the ranks was applied, with the Dunn's posthoc test. We rejected the null hypotheses (i.e. that there are no differences

between the means of the groups) at the 0.05 significance level. We used R (R Core Team, 2017) with the "PMCMRplus" package (Pohlert, 2018) in R to perform the non-parametric comparisons.

Next, we analyzed the overall effect of natural infrastructure interventions on soil erosion and soil quality indicators. In this analysis, we only included case studies with a control–treatment design, where quantitative measures of soil erosion and quality

were available to establish the control–treatment contrast. The response ratio (RR) was then used to determine the effect sizes.

In this study, the response ratio was calculated for each natural infrastructure intervention (PRO, FOR, SWC) and soil erosion and quality indicator (Sloss, RC, SSY, RCC, SOC, BD). For the control group, we combined data of sites with traditional agriculture, either cropland (CROP) or rangeland (RANGE), and bare land (BARE) given the limited number of matched pairs of control and single or multiple treatment(s). For each pairwise comparison, we plotted the effect size of the individual studies

in forest plots and explored the heterogeneity in the response among the case studies. These plots were used to identify the magnitude and sources of variation among the studies, and to identify eventual outliers. We then extracted the central tendency (mean effect) and confidence limits (standard error) for each indicator and pairwise comparison. The mean effect and its standard error were plotted in summary forest plots (per pairwise comparison) to assess the overall effectiveness of a specific intervention on soil erosion and quality indicators. The graphs were produced using the R-package "metafor" (Viechtbauer,

2010).

**LANDSCAPE ELEMENTS**

| Natural vegetation | Traditional Agriculture | | Bare land |
|---|---|---|---|
| Forests and Native grasslands | Cropland (CROP) | Rangeland (RANGE) | (BARE) |

**TYPE OF INTERVENTION**

| Restoration and Protection (PRO) | Forestation (FOR) | Soil and Water Conservation Measures (SWC) | None |
|---|---|---|---|

**INDICATORS**

| On-site Soil Erosion | Off-site Soil Erosion | Soil quality (topsoil) |
|---|---|---|
| Soil Loss Rate [t.km$^{-2}$.yr$^{-1}$] (Sloss) | Sediment yield [t.km$^{-2}$.yr$^{-1}$] (SSY) | Soil Organic Carbon [%] (SOC) |
| Plot Runoff Coefficient [%] (RC) | Runoff Ratio [% ] (RCC) | Bulk Density [g.cm$^{-3}$] (BD) |

**Figure 2. Schematic overview of study design**

# 3 Results and discussion

## 3.1 Overall descriptive statistics

Of the 118 studies evaluating the effect of natural infrastructure interventions on soil erosion and quality indicators, 54 studies contained data on soil and water conservation practices (SWC), 50 studies on protective vegetation (PRO, FOR or both) and 14 studies on all three of them (SWC, PRO and FOR). The majority of studies were journal articles (79 %), followed by grey literature (14 %) and chapters from books (7 %). The studies cover a 6500-km long stretch across the Andes, with 4 % of the studies in Venezuela (n = 5), 6 % in Colombia (n = 7), 36 % in Ecuador (n = 43), 35 % in Peru (n = 41), 7 % in Bolivia (n = 8), 8 % in Chile (n = 9) and 4 % in Argentina (n = 5). Ecuador and Peru have the highest concentration of case studies (Fig. 3). The large majority, i.e. 89 %, of the studies have investigated soil erosion in tropical climates, with 59 % of the studies performed in pluvial seasonal, 19 % in pluvial and 10 % in desertic or xeric climates. The remaining studies were performed in temperate or Mediterranean climate regimes. Field studies mostly involved erosion measurements at the plot scale (48 %, n = 57), small catchment scale (18 %, n = 21), and landscape scale (22 %, n = 26). Only 12 % of the studies included erosion assessment at the scale of large catchments (> 1000 km$^2$).

**Table 1.** Summary of the mean indicator values per treatment, with indication of the number of individual case studies between brackets. The values are reported for three interventions in natural infrastructure : PRO = restoration and protection of natural vegetation like forest or native grasslands, FOR = forestation with native and/or exotic species, and SWC = implementation of soil and water conservation measures, and for three untreated areas : CROP = cropland, RANGE = rangeland under traditional agricultural management, and BARE = bare land corresponding to abandoned cropland or degraded land with very low (< 10 %) vegetation cover. Difference between groups was tested with the Kruskal-Wallis rank sum test, and p-values are estimated using the chi-square distribution. Means followed by a common letter are not significantly different by the Dunn's non-parametric all-pairs comparison test at 5% level of significance.

| | | SOC | BD | Sloss | RC | SSY | RCC |
| --- | --- | --- | --- | --- | --- | --- | --- |
| | | [%] | [g cm$^{-3}$] | [t km$^{-2}$ yr$^{-1}$] | [%] | [t km$^{-2}$ yr$^{-1}$] | [%] |
| Treatment | PRO | 8.67$^a$ (16) | 0.82$^a$ (12) | 287$^a$ (10) | 14.0$^{ab}$ (2) | 1095 (10) | 35.7 (6) |
| | FOR | 3.17$^{ab}$ (8) | 1.05$^{ab}$ (4) | 1860$^{ab}$ (5) | 15.7$^{ab}$ (1) | 1405 (7) | 23.8 (4) |
| | SWC | 2.96$^b$ (15) | 1.37$^b$ (6) | 1660$^{ab}$ (39) | 6.40$^b$ (9) | 1883 (4) | 36.1 (2) |
| Control | RANGE | 6.21$^{ab}$ (17) | 1.10$^{ab}$ (12) | 2370$^{ab}$ (14) | 24.5$^{ab}$ (2) | 464 (3) | 41.0 (2) |
| | CROP | 2.49$^b$ (19) | 1.02$^{ab}$ (6) | 3250$^b$ (39) | 6.80$^{ab}$ (10) | 3417 (5) | 38.5 (2) |
| | BARE | 1.88$^b$ (10) | 1.23$^{ab}$ (6) | 5140$^b$ (16) | 20.0$^a$ (13) | 6170 (8) | 53.1 (2) |
| | ALL | | | | | | |
| | $\bar{x} \pm 1SE$ | 4.47 ± 0.62 | 1.07 ± 0.05 | 2590 ± 295 | 12.9 ± 2.23 | 2600 ± 570 | 35.9 ± 4.54 |
| | n (#) | 85 | 46 | 123 | 37 | 37 | 18 |
| | *p-value* | *<0.01* | *0.02* | *<0.01* | *0.03* | *0.10* | *0.59* |

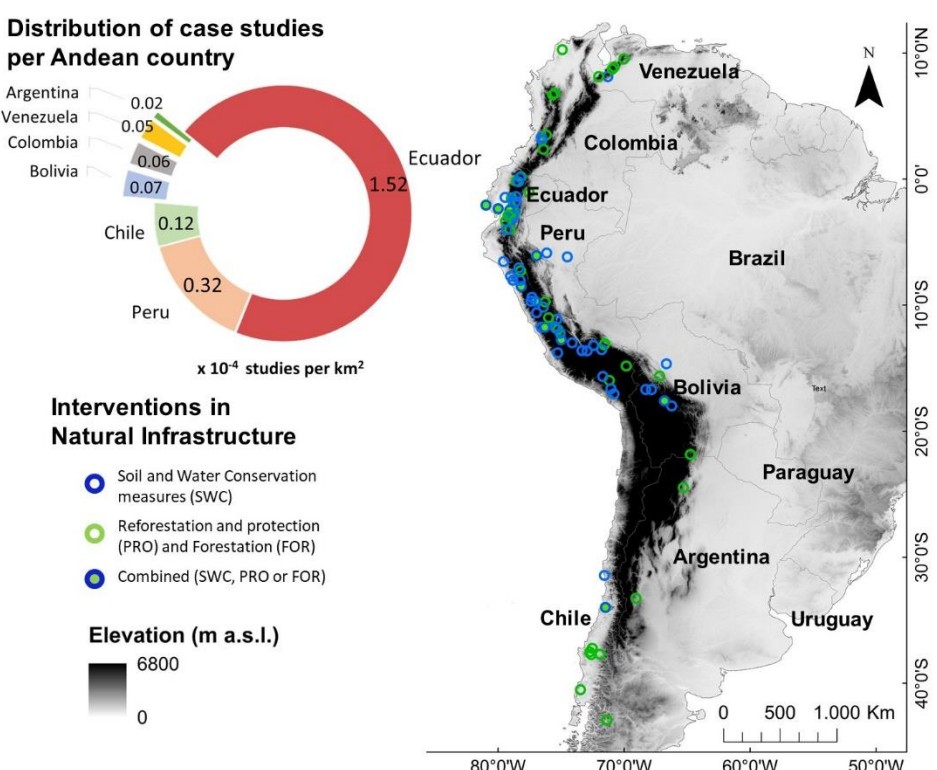

**Figure 3.** Spatial distribution of case studies in the Andean region, classified per type of natural infrastructure intervention. The background map corresponds to the 30 arc-second DEM of South America (GTOPO30, U.S. Geological Survey's Center for Earth Resources Observation and Science, EROS).

### 3.2 Erosion mitigation assessed from different soil erosion indicators

The one-way analysis of variance revealed significant differences between treatment and control in soil quality and on-site soil erosion with notable differences in soil organic carbon ($p < 0.01$, $n = 85$), bulk density ($p = 0.02$, $n = 46$), soil loss ($p < 0.01$, $n = 123$) and plot runoff coefficient ($p = 0.03$, $n = 37$) (Table 1; Fig. 4). Notably, none of the erosion indicators that were measured at the catchment scale were significant at the 0.05 level, as we observe only small differences between categories for specific sediment yield ($p = 0.10$, $n = 37$) and no differences for catchment-wide runoff coefficient ($p = 0.59$, $n = 18$). The latter might be due to the limited number of observations documenting the effect of natural infrastructure interventions on SSY ($n = 37$) or RCC ($n = 18$) and inherent variability in runoff and sediment discharge at the catchment scale as shown by Tolorza et al. (2014) and Molina et al. (2015). Below, we only present tendencies that are statistically significant at the 0.05 level.

Soil organic carbon concentration of topsoil is the indicator with the highest significance level, showing strong differences in soil quality between protected sites, cropland and bare soil (Table 1, Fig. 4). Based on 85 observations, we observe soil organic carbon concentrations of the topsoil between 0.47 % and 34.06 %, with mean values of 4.47 ± 0.62 %. Based on the results of the posthoc Kruskal-Wallis rank sum test, two distinct groups can be identified: (1) areas covered by natural vegetation such as forests and native grasslands with a mean SOC value of 8.67 ± 1.89 %, and (2) areas covered by agricultural crops and bare land having a mean SOC value of, respectively, 2.49 ± 0.32 % and 1.88 ± 0.49 %. Areas with soil and water conservation measures belong to the 2$^{nd}$ group with a SOC of 2.96 ± 0.70 %. Rangelands and plantation forests have intermediate SOC values of resp. 6.21 ± 2.05 %, and 3.17 ± 0.71 %. These results are consistent with the systematic review of Bonnesoeur et al. (2019) that reported lower levels of topsoil organic matter in plantations compared to native forests and grasses. However, the differences reported here are not statistically significant (p=0.12).

Soil bulk density of the topsoil is reported in 15 % of the reported case studies for different natural infrastructure interventions. Soil bulk density ranges between 0.36 and 1.67 g cm$^{-3}$ with a mean value of 1.07 ± 0.05 (Fig. 4; Table 1). The lowest mean BD values, i.e. 0.82 ± 0.08 g cm$^{-3}$, are observed in soils covered by native vegetation. Although the mean BD values are notably higher in areas with cropland, forestation, rangeland, and particularly bare land, the wide range in reported BD values per category does not allow us to distinguish them from areas covered by natural vegetation at the 0.05 significance level. Remarkably, areas under soil and water conservation treatments have significantly higher bulk density values compared to natural vegetation (Table 1), which might reflect the advanced state of physical soil degradation due to compaction before SWC intervention (e.g. Rymshaw et al., 1997). It also highlights that it may take several years to decades for impacts to be reversed, and that high levels of subsurface compaction may be irreversible without soil restoration (Borja, 2018).

The rate of soil loss measured at the plot scale (t km$^{-2}$ yr$^{-1}$) is one of the most common indicators of soil erosion, as it is reported in 43 % of the case studies. The 125 quantitative measurements of Sloss reveal that soil loss rates vary widely with mean value of 2590 ± 295 t km$^{-2}$ yr$^{-1}$ and minimum and maximum values of resp. 0.001 and 14761 t km$^{-2}$ yr$^{-1}$. Significant differences in Sloss are observed between areas covered by natural vegetation and crop- or bare land, with soil losses being – on average – 11 to 18 times lower in areas with protected vegetation (Table 1). Rangelands, areas with forestation and soil and water conservation measures have intermediate values of Sloss (resp. 2370, 1860 and 1660 t km$^{-2}$ yr$^{-1}$), and are not significantly different from natural vegetation, crop- or bare land.

The runoff coefficient (RC) is measured as surface runoff at the plot scale, and is here reported as the percentage of the rainfall that becomes runoff. The number of case studies that report runoff coefficients for different categories of natural infrastructure is low (12 %). Figure 4 illustrates the wide range of RC values (min: 0 %, max: 47 %) that are observed in the Andes, with mean values of 12.9 ± 2.23. The large variation might be the result of inherent spatial heterogeneity in rainfall-runoff response (Guzman et al., 2019). However, methodological bias cannot be excluded as multiple field methods to estimate plot runoff

coefficients were used: portable rainfall simulators covering a few cm² (e.g. Harden, 2001), runoff plots covering 1 m² (e.g. Perrin et al., 2001; Molina et al., 2007), and experimental sites covering > 10 m² (Molina et al. 2009; Suescún et al. 2017).

Also, the amount and intensity of the (simulated) rainfall often vary between case-studies. Notwithstanding, significant differences are observed in RC between areas with soil and water conservation measures and bare land, with RC values being, on average, 3.5 times lower in SWC compared to BARE (Fig. 4).

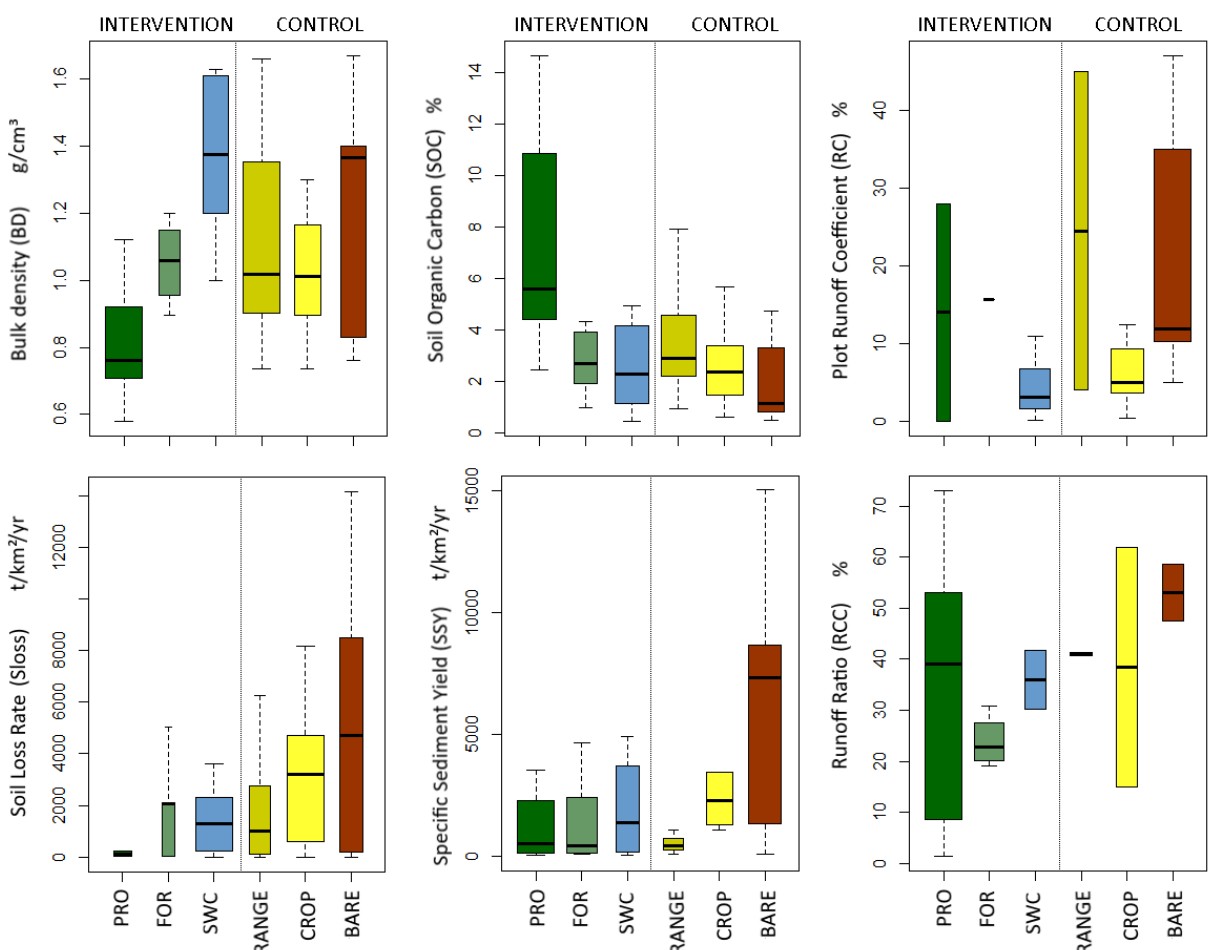

**Figure 4.** Variation in bulk density, soil organic carbon content, runoff coefficient, soil loss rate, specific sediment yield and
305 catchment runoff coefficient between the three categories of natural infrastructure intervention (PRO, FOR, SWC), the untreated agricultural (RANGE, CROP) and bare land (BARE). Bold lines represent the median values, boxes extend to first and third quantiles and whiskers to 1.5 times the interquartile range from the box.

**3.3 Reported effectiveness of natural infrastructure interventions on soil erosion and soil quality**

When limiting the quantitative analysis to matched pairs of control and single or multiple treatment(s), the number of independent empirical studies is reduced from 118 to 89 studies. For the analysis of the response ratios, the sites with traditional agriculture, either cropland (CROP) or rangeland (RANGE), and bare land (BARE) were regrouped into one control group. This is justified by the fact that the soil quality and erosion indicators are not significantly different between the three types of control sites (Table 1). Figure 5 and Table 2 show the effect size of (i) restoration and protection of natural vegetation, (ii)

forestation, and (iii) soil and water conservation on soil organic carbon, bulk density, runoff coefficient, soil loss rate, specific sediment yield and catchment-wide runoff coefficient. Below, we only discuss results that are based on a minimum of 4 independent treatment-control studies.

**Table 2.** Summary of response ratios, showing the effect of restoration and protection of natural vegetation (PRO-, forestation

(FOR) and soil and water conservation (SWC) on soil quality and erosion. The mean value and the 68% confidence interval are given, as well as the number of treatment-control studies. When the number of studies is below 5, the reported values are shown in light grey.

| Effect size on | Restoration and Protection of Natural Vegetation (PRO) | | Forestation (FOR) | | Soil and Water Conservation (SWC) | |
|---|---|---|---|---|---|---|
| | Mean (68% CI) | # | Mean (68% CI) | # | Mean (68% CI) | # |
| SOC | 3.01 (2.12-3.90) | 26 | 1.19 (1.06-1.31) | 12 | 1.28 (1.11-1.45) | 17 |
| BD | 0.878 (.848-.908) | 16 | 0.959 (.926-.991) | 6 | 0.946 (.914-.978) | 9 |
| RC | 0.318 (.159-.477) | 4 | 1.20 | 1 | 0.740 (.424-1.06) | 16 |
| Sloss | 0.357 (.170-.545) | 14 | 1.95 (.922-2.98) | 7 | 0.621 (.529-.714) | 51 |
| SSY | 0.334 (.250-.419) | 10 | 0.713 (.436-.990) | 8 | 3.88 (.399-7.35) | 3 |
| RCC | 0.830 (.650-1.01) | 3 | 0.532 (.416-.649) | 2 | 0.774 (.513-1.03) | 2 |

Amongst the three intervention types, the protection and conservation of natural vegetation (PRO) has the strongest effect on soil quality (SOC, BD) and erosion (Sloss, SSY). When native forests and grasses are protected from conversion to agricultural land, the topsoil contains $3.01 \pm 0.893$ times more soil organic carbon than in the control sites. At the same time, the soil physical structure is better with a dry bulk density of $0.82 \pm 0.08$ g cm$^{-3}$ in natural vegetation, being $0.878 \pm 0.030$ times lower compared to control sites. The high soil porosity enhances structural support, water and solute movement and soil aeration

(Podwojewski et al., 2002). There is a clear and positive effect on soil erosion mitigation, with soil losses being $0.357 \pm 0.187$

times lower, and specific sediment yields being $0.334 \pm 0.085$ times lower than at the control sites. Experimental work by e.g. Janeau et al. (2015) showed the importance of native vegetation in facilitating soil water infiltration as it can conduct over 50% of rainwater through stemflow to the soil. This is confirmed by other empirical data (e.g., Harden, 1996; Poulenard et al., 2001) collected at the plot scale, and the runoff coefficient is – on average – $0.318 \pm 0.159$ times lower in areas where natural vegetation is protected or conserved. The empirical data is not sufficient to assess systematically the effect on runoff processes at the catchment scale.

Only 17% of the records on treatment-control experiments contain information on the effect of forestation with native and/or exotic species (FOR) on soil quality, on-site and off-site soil erosion (Table 2). The database counts less than 3 empirical studies on rainfall-runoff generation, at the plot and catchment scale. Compared to the control sites, a positive effect is reported on soil quality, with $1.19 \pm 0.125$ times higher SOC and $0.959 \pm 0.032$ lower bulk density. The pairwise analysis did not show evidence of a net effect of forestation on soil erosion (Sloss): the response ratio shows large scatter with RR values ranging between 0.37 in the study by Henry et al. (2013) and 7.75 in the case published by Pesantez and Seminario (2010). Notwithstanding the high variability in response ratios for on-site erosion (Sloss), a positive effect was observed for the catchment-wide sediment yields with SSY being –on average– $71.3 \pm 27.7$ % of the yields measured in control sites. Similar observations were made by Bonnesoeur et al. (2019) who attributed the scatter in the empirical studies to the type of forestation (native vs. exotic species) and forestation age. In addition to this, the prior state of the environment (soil quality and erosion) has a major impact on erosion mitigation as Balthazar et al. (2015) showed for a case in the Ecuadorian Andes.

Almost 50% of the treatment-control studies concern interventions with soil and water conservation measures (SWC). There is a net positive effect of the implementation of conservation measures on the soil organic carbon content of the topsoil, with values that are $1.28 \pm 0.170$ times higher compared to control sites (Fig. 5). The effect on the bulk density is small, with BD in treated sites being $94.6 \pm 3.2$ % of the values measured in control sites. The limited effect on soil bulk density suggests that the recovery of the soils' physical structure from compaction is slow, even within the topsoil (Jacobi et al., 2015). Soil loss rates changed significantly after the application of soil and water conservation measures: soil losses are reduced to $62.1 \pm 9.2$ % of their original values after the intervention. The effect of infiltration ditches on soil erosion mitigation is particularly well documented for the Peruvian Andes, where Vasquez and Tapia (2011) reported soil erosion rates that were more than two times reduced after the intervention (i.e., from 4500 to 2060 t km$^{-2}$ yr$^{-1}$). The effect is strongest when the measures are applied on abandoned cropland or degraded land with very low (< 10 %) vegetation cover (Fig. 3), as shown by De Noni et al. (2001) in various case-studies distributed along the Ecuadorian Andes. In contrast to the response ratios for soil loss, the plot-scale runoff coefficient shows large scatter with values of $0.740 \pm 0.316$. The scatter can be attributed to strong differences in hydrological response between control sites, with degraded and abandoned land generating more runoff than rangeland or arable land (Molina et al., 2007).

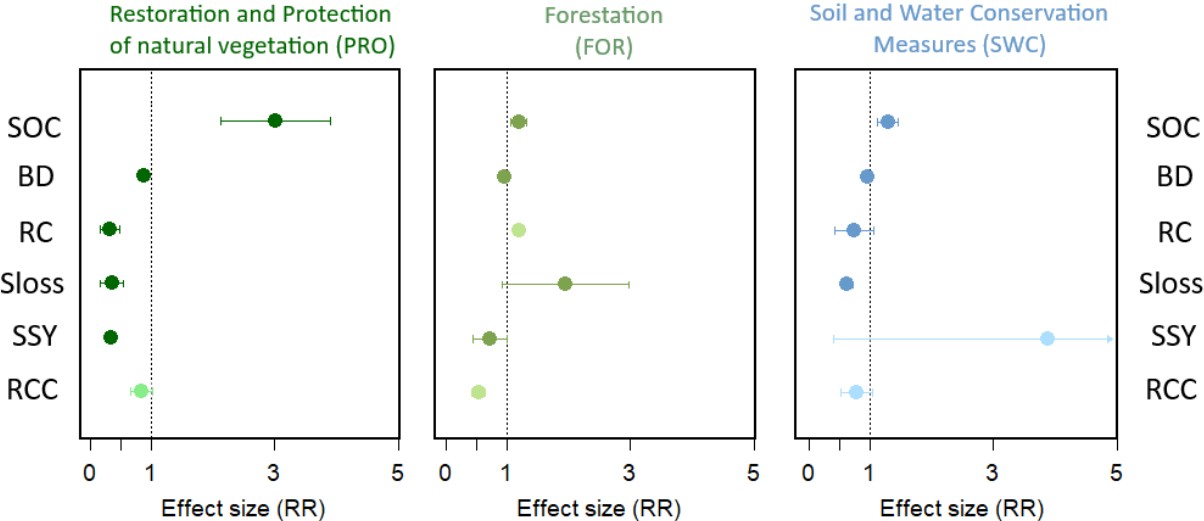

Effect of specific intervention type as measured in matched pairs of control and treatment(s)

**Figure 5.** Response ratio of natural infrastructure interventions (PRO, FOR, SWC) and untreated degraded land (RANGE, BARE) relative to cropland and rangeland. The plots show the mean response ratio (points) and its standard error (solid lines) for soil organic carbon (SOC) and bulk density (BD) in the topsoil, runoff coefficient (RC) and soil loss rate (Sloss), specific sediment yield (SSY) and catchment-wide runoff ratio (RCC). When the number of individual treatment-control studies is below 4, the symbols are shown in lighter colours.

### 3.4 Knowledge gaps and prospects for future research

### 3.4.1 Representation of natural variability in environmental conditions within the Andean region

The literature reviewed in this study showed an unequal distribution of empirical studies over the Andean countries, with an under-representation of studies from Argentina, Venezuela, Colombia and Bolivia (Fig. 3). Grey literature (e.g., technical reports) from these countries was often inaccessible via standard search methods, in contrast to grey literature from Peru or Ecuador. When compared to the Andean region, the dataset of 118 case-studies contains a disproportionally high amount of studies from mid elevations (i.e., between 2000 and 4000 m a.s.l.) and moderate relief with hillslope gradients below 15° (Fig. 6). High elevation sites, and areas with either low or high relief are underrepresented. Similarly, the regions with intermediate precipitation amounts are overrepresented, and there is a disproportionally low amount of studies with either low to very low (< 400 mm per year) or very high (> 3000 mm.yr$^{-1}$) precipitation (Fig. 6). Given the spatial bias in the data compilation, the records do not allow to perform a statistically unbiased regional-scale assessment of water erosion mitigation. It is necessary that future studies collect empirical data on soil quality, erosion and sediment yield before/after interventions in the above-mentioned data-scarce regions.

There is a particular lack of knowledge on soil erosion processes before, after or during extreme rainfall or seismic events. Of the 118 quantitative studies, only 20 studies or 17 % explicitly referred to flooding or erosion processes during extreme (i.e. high-magnitude but rare and episodic) events. Reliable, quantitative information about the return period of extreme erosion and flooding events, and their influence on soil quality, long-term erosion rates and sediment discharge is scarce (Aguilar et al., 2020; Carretier et al., 2018). The severe scarcity of studies on the impact of extreme events has major implications for

informing land use management practices (Coppus and Imeson, 2002), as the effectiveness of policy-based interventions on natural infrastructure could not be methodically evaluated for extreme events. A number of model applications by e.g. Bathurst et al. (2011, 2020) conveyed the limitations of forestation as an intervention for reducing peak discharges of floods derived from extreme but infrequent rainfall events. There is a clear need to thoroughly evaluate whether our results on the effectiveness of natural infrastructure interventions during frequent erosion events can be extended to extreme events related

to e.g. El Niño–Southern Oscillation (ENSO).

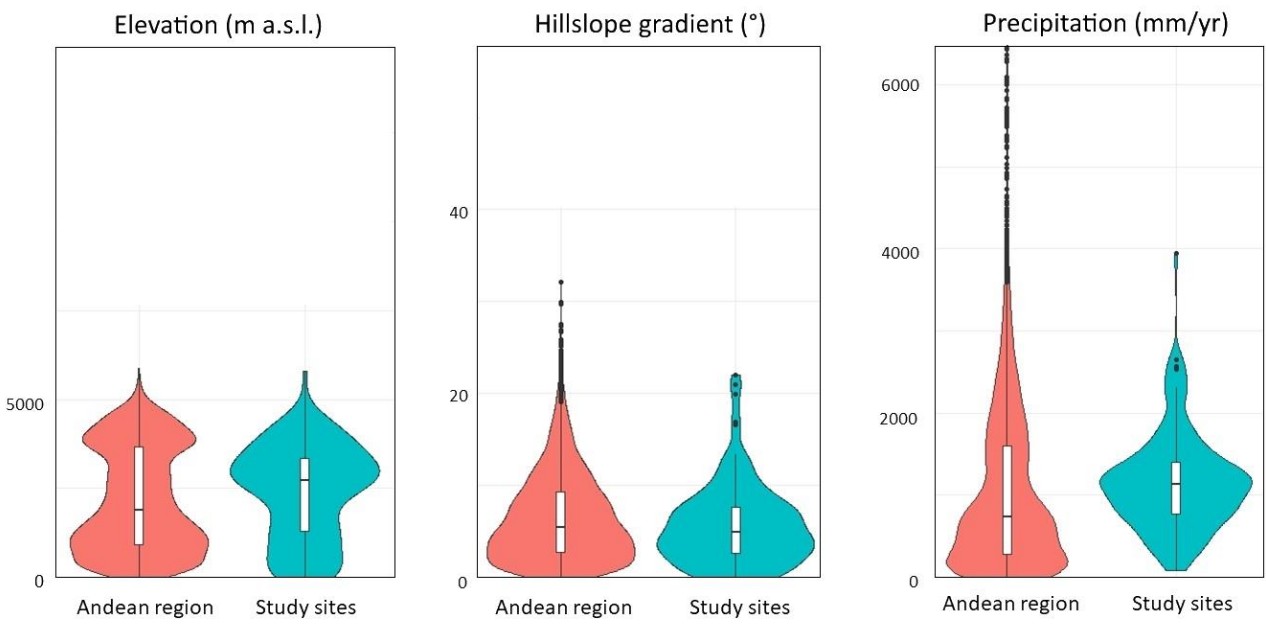

**Figure 6:** Distribution of the mean elevation (m a.s.l.), mean annual precipitation (mm yr$^{-1}$) and mean hillslope gradient (°) for the 118 empirical studies and the entire Andean region. The delineation of the Andean region is based on Körner et al.

(2016). The topographic information is derived from the 30 arc-second DEM of South America (GTOPO30, U.S. Geological Survey's Center for Earth Resources Observation and Science, EROS), and the mean annual precipitation data from the Tropical Rainfall Measuring Mission (TRMM 3B43) dataset (Ceccherini et al., 2015).

### 3.4.2 Gap between plot-scale and catchment-scale erosion assessments

There is a clear gap between the number of case studies on water erosion at the plot-scale and the catchment-scale with about 48 % of all articles on plot-scale erosion phenomena, and only 30 % on sediment yield at small and large catchment scale. The remaining 22 % of the studies are conducted at landscape scale, with observations made at different topographic positions within a larger geographical region. Due to their replicability, erosion plot studies are the most used and standardised experimental method, whereby runoff and sediment are measured from bounded runoff plots of $\leq 1$ m² (e.g., Harden, 1996;

Poulenard et al., 2001) to 1000 m² (e.g., De Noni et al., 2001). The strong focus on soil erosion mitigation on farmers' land is in line with past and ongoing efforts on sustainable and resilient agriculture by programs like PRONAMACHCS and MARENASS in Peru, PRONAREG-MAG-ORSTOM and USAID in Ecuador and IIDE and USAID in Bolivia. Even when local interventions have proven very successful, they are seldomly implemented at a large scale. Only a handfull of studies (e.g., Molina et al., 2007; 2008) in our database evaluated water erosion simultaneously at the plot and catchment scales.

Therefore, it is necessary that future studies are designed to assess the effectiveness of water-related interventions on the specific sediment yield or catchment-wide runoff ratio at the broader catchment scale.

Transferring knowledge on erosion mitigation from the plot-scale to the catchment-scale remains a challenge. First, local-scale erosion phenomena might not be representative for the dominant erosion processes at the catchment scale. For example, while

farming terraces or infiltration ditches enhance infiltration and reduce runoff and erosion on hillslopes (e.g., Sandor and Eash, 1995), localised sediment sources such as runoff generating unpaved roads or debris flows might overwhelm the sediment yield (e.g., Vanacker et al., 2007b). This can also be observed by the divergence in response ratios of soil loss rates and specific sediment yields, after implementation of soil and water conservation measures (Fig. 5). Second, when the sediment that is generated by water erosion on the hillslopes is transferred downslope to the river network, sediment storage, erosion and

remobilisation can occur across the river system (Romans et al., 2016). The effect of a specific intervention (like forestation) on soil erosion on the hillslopes is therefore not directly leading to a similar change in sediment yield at the outlet of the catchment (Fig. 5). Further empirical work is needed to decipher how environmental signals, such as changes in erosion rates after natural infrastructure interventions, are transferred through hillslopes, floodplains and river channels.

**4 Conclusion**

The systematic review of grey and peer-reviewed literature on natural infrastructure interventions and erosion mitigation in the Andean region resulted in 1798 potentially relevant case studies. After screening the records, 118 empirical studies were eligible and included in the quantitative analysis on soil quality and soil erosion. From the suite of physical, chemical and biological indicators commonly used in soil erosion research, six indicators were pertinent to study the effectiveness of natural

infrastructure: soil organic carbon and bulk density of the topsoil, soil loss rate and runoff coefficient at the plot scale, and specific sediment yield and catchment-wide runoff coefficient at the catchment scale. The one-way analysis of variance

revealed significant differences between treatment and control in soil organic carbon ($p < 0.01$, $n = 85$), bulk density ($p = 0.02$, $n = 46$), soil loss ($p < 0.01$, $n = 123$) and plot runoff coefficient ($p = 0.03$, $n = 37$). None of the erosion indicators that were measured at the catchment scale were significant at the 0.05 level.


The protection and conservation of natural vegetation has the strongest effect on soil quality and erosion. When native forests and grasses are protected from conversion to agricultural land, the topsoil contains $3.01 \pm 0.893$ times more soil organic carbon, and has a better physical structure than the control sites. At the same time, there is a clear effect on erosion mitigation, with soil losses being $0.357 \pm 0.187$ times lower, and specific sediment yields being $0.334 \pm 0.085$ times lower than at control sites.

The effect of forestation with native and/or exotic species is less documented. A positive effect is reported on soil quality, with $1.19 \pm 0.125$ times higher SOC and $0.959 \pm 0.032$ lower bulk density compared to control sites. The pairwise analysis did not show evidence of a net effect on soil erosion, although a positive effect was observed for catchment-wide sediment yields being $71.3 \pm 27.7$ % of the yields measured in control sites. The implementation of soil and water conservation measures has a net positive effect on the soil organic carbon content of the topsoil, with values that are $1.28 \pm 0.170$ times higher than control

sites; and on soil loss rates that are reduced to $62.1 \pm 9.2$ % of their original values after the intervention.

The systematic review of the existing literature allowed us to identify critical gaps in knowledge and research. We observed spatial bias in the data compilation. There is a need for future empirical work on soil quality, erosion and sediment yield before/after interventions in data-scarce regions, such as high elevations, regions with either low or high relief, and low to very

low or very high precipitation. Besides, most erosion assessments are based on short-term measurements that tend to miss the impact of rare, high-magnitude events. It is necessary to evaluate whether the results of this study on the effectiveness of natural infrastructure interventions hold during extreme events related to e.g. El Niño–Southern Oscillation (ENSO). In addition, future climate variability and global warming might trigger erosion events, as freshly exposed deglaciated terrain is particularly prone to soil erosion.


**Supplements**

Supplement A. Definitions of terms used in the systematic review (in English and Spanish)

Supplement B. List of databases that were searched, with indication of search terms

Supplement C. Structure of the database

Supplement D. List of the literature studies that were included in the systematic review

Supplement E. Test library of 20 references compiled by experts in the fields

**Data availability**

The data underpinning this study have been archived with PANGAEA (link). The data may also be requested from the corresponding author by email.

**Author contribution**

VV and AM conceived the study and conducted the statistical analyses, with backstopping of VB, FRD and BOT. AM and MR compiled and accurated the database from peer-reviewed and grey literature. All authors contributed to shaping the research and analyses, as well as writing the paper.

**Competing interests**

The authors declare that they have no conflict of interest.

**Funding information**

We acknowledge funding from the Natural Infrastructure for Water Security Project funded by USAID and the Government
of Canada. BOT also acknowledges the National Secretariat of Higher Education, Technology and Innovation of Ecuador (SENESCYT). MR was supported by a PhD Scholarship from the Conseil de l'Action Internationale from UCLouvain (Grant No. ADRI/CD/CA/2016-NR 51) and the Académie de Recherche et Enseignement Supérieur de la Fédération Wallonie-Bruxelles (ARES) Belgium. WB received funding from the UKRI Natural Environment Research Council (contract NE/S013210/1).

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
