# Peer review of "The effect of natural infrastructure on water erosion mitigation in the Andes"

_SOIL, 2021_

## Referee Comment (RC1)

**For Authors**

**"The effect of natural infrastructure on water erosion mitigation in the Andes"**

This study shows interesting results through a systematic review of 121 case-studies from the Andes to define useful "erosion indicators" i) Effectiveness of natural infrastructure? Ii) Impact of working with natural infrastructure on on-site and off-site erosion mitigation? Iii) locations and types of studies necessary to complete knowledge and research.

- Weak point: some information or references about sites/countries and rain characteristics are unclear or missing because the research of the studies is done by a specific selection of key words.
- Strong point: the overview of the selected references is very well analysed.

The document could be improved following the comments below:

➢ **Introduction**

Line 54: a large research and applied project was conducted in Ecuador to protect Quito City, cf. *Perrin JL, Bouvier C, Janeau JL, Menez G, Cruz F. Rainfall/runoff processes in a small peri-urban catchment in the Andes mountains. The Rumihurcu Quebrada, Quito (Ecuador). Hydrological processes 2001; 15: 843-854.*

I noted that you mentioned this study in the supplement D.

➢ **Materials and methods**

Line 102 then line 110 to 115 and Supplement C:

I can understand that you selected studies due to absence of quantitative on-site or off-site soil erosion or soil quality measurements and I know that it is difficult to collect all the studies carried out on your topic, however it seems that some studies, are missing despite using your key words:

*land use*, *Ecuador, infiltration*, *soil erosion*

Buytaert W, Wyseure G, De Bievre B, Deckers J. The effect of *land-use* changes on the hydrological behaviour of Histic Andosols in south Ecuador. Hydrological processes 2005; 19: 3985-3997.

Janeau JL, Grellier S, Podwojewski P. Influence of rainfall interception by endemic plants versus short cycle crops on water *infiltration* in high altitude ecosystems of Ecuador. Hydrology Research 2015; 46: 1008-1018.

Poulenard J, Podwojewski P, Janeau JL, Collinet J. Runoff and *soil erosion* under rainfall simulation of Andisols from the Ecuadorian Paramo: effect of tillage and burning. Catena 2001; 45: 185-207.

**Data base development – line 161 to 170:**

In this paragraph, I appreciated the definition of the soil indicators, but I suggest providing more explanation for the role of the rain (different intensities and duration depending on altitude…) and the role of the vegetal cover (throughfall/stemflow for infiltration, soil protection by different type of covers…), the broad range of bulk density (type of volcanic ashes).

This complement of explanation could be useful to explain your result Line 284 to 291 in **Results and Discussion** and your Figure 4.

➢ **Results and Discussion**

**3.4.1 Representation of natural variability in environmental conditions within the Andean region**

The runoff and erosion processes decrease significatively (excepted exceptional events) with appropriate technique to protect the soil (cultural, terraces, etc.), with vegetal cover (natural or anthropized origin).

This is not a surprise, and your review shows that what has already been studied on these "parameters".

However, the rainfall characteristics (intensity, concentration of precipitation, etc.) seem little mentioned. Is it correct?

**3.4.2 Gap between plot-scale and catchment-scale erosion assessments,** apparently, we have a "confusion" about erosion processes depending on the studied scale.

✓ Micro-plots $\leq 1$ m$^{-2}$ to 10 m$^{-2}$, generally the authors are studying Detachability and/or Erodibilty expressed in gramme per m$^{-2}$

✓ Catchment scale the authors are interested by erosion expressed in tons per hectare.

Due to this difference in scale, it is therefore very difficult to compare the studies with each other.

Can you be more specific on how you compare these two spatial scales?

➢ **Conclusion and perspectives.**

It is a long conclusion about the contains of this review, maybe too long. On the other hand, the research perspectives look few and insufficiently precise.

Could you add/suggest some specific studies should be done on the rainfall intensity thresholds corresponding to the effect of soil erosion parameter. (Which ones? How?)?

Is it desirable to mention more studies related to climate change?

➢ **Figure 3**

A small green circle is in Argentina but apparently far away of Andes mountains. Is it really related with your keywords and/or scientific questions?

---

## Author Comment (AC1)

**Point by point response to Referee#1 comments**

Dear Referee#1,

We would like to thank you for the insightful comments on our manuscript "**The effect of natural infrastructure on water erosion mitigation in the Andes »** (**soil-2021-76). We are pleased to read that you appreciated the systematic review. We take your point that the study can be improved by better defining erosion/erodibility parameters like climate, vegetation, parent material; and by addressing how erosion rates are compared over a wide range of spatial scales. We take note of the references, and will include them in the revised paper.

Please find in the supplementary file a point-by-point response to all the concerns raised and how we will address them. Your original comments are copied below, and we have highlighted in blue color how we will revise the manuscript accordingly. We hope you find our response and changes to the manuscript satisfying and we are looking forward to hearing from you.

With kind regards,
Veerle Vanacker on behalf of all co-authors

**Major comments**
- Weak point: some information or references about sites/countries and rain characteristics are unclear or missing because the research of the studies is done by a specific selection of key words.
- Strong point: the overview of the references is well analysed.
*We acknowledge that we could have included more information about the overall characteristics of the study sites, like rainfall amounts or intensities. We will provide a summary with an overview of site characteristics in the revised paper.*

With some improvements, especially verifying the interest of specific references, in the definition of the erosion/erodibility parameters, and the conclusion-perspectives this paper could provide a relevant review for the research community working on different aspect of Andes mountains "ecology" (interaction soil, water, vegetation) and more generally for erosion processes studies.
*Thank you for the suggestions. We are familiar with these studies, as they were part of the full papers that we assessed in this study. In the revised manuscript, we will add more information on the quantification of the erosion rates, and the erodibility parameters.*

**Specific comments**
Line 54: a large research and applied project was conducted in Ecuador to protect Quito City, cf. *Perrin JL, Bouvier C, Janeau JL, Menez G, Cruz F. Rainfall/runoff processes in a small peri-urban catchment in the Andes mountains. The Rumihurcu Quebrada, Quito (Ecuador). Hydrological processes 2001; 15: 843-854.* I noted that you mentioned this study in the supplement D.

*We thank the referee for this suggestion, and we agree that we can refer to this study in the introduction where we refer to studies analyzing the effect of land use on erosion processes.*

Line 102 then line 110 to 115 and Supplement C:

I can understand that you selected studies due to absence of quantitative on-site or off-site soil erosion or soil quality measurements and I know that it is difficult to collect all the studies carried out on your topic, however it seems that some studies, are missing despite using your key words:

Buytaert W, Wyseure G, De Bievre B, Deckers J. The effect of **land-use** changes on the hydrological behaviour of Histic Andosols in south Ecuador. Hydrological processes 2005; 19: 3985-3997.

Janeau JL, Grellier S, Podwojewski P. Influence of rainfall interception by endemic plants versus short cycle crops on water **infiltration** in high altitude ecosystems of Ecuador. Hydrology Research 2015; 46: 1008-1018.

Poulenard J, Podwojewski P, Janeau JL, Collinet J. Runoff and **soil erosion** under rainfall simulation of Andisols from the Ecuadorian Paramo: effect of tillage and burning. Catena 2001; 45: 185-207.

*We are aware of these studies, as they were part of the peer-reviewed studies that we initially retrieved from SCOPUS. The three studies are highly relevant for studying rainfall-runoff processes in the Andes. In the screening phase, we removed studies that did not contain quantitative data on soil erosion (soil erosion rates measured as $M/L^2/T$ or $L/T$). Thereby, the database was reduced to 121 studies.*

**Data base development – line 161 to 170:** In this paragraph, I appreciated the definition of the soil indicators, but I suggest providing more explanation for the role of the rain (different intensities and duration depending on altitude…) and the role of the vegetal cover (throughfall/stemflow for infiltration, soil protection by different type of covers…), the broad range of bulk density (type of volcanic ashes). This complement of explanation could be useful to explain your result Line 284 to 291 in Results and Discussion and your Figure 4

*We thank the referee for this suggestion, and will include more information on the factors controlling erodibility and erosivity, like vegetation cover, parent material and rainfall. Part of the variability in response ratios (shown in Figure 4) can be attributed to these factors, and we will further develop this in the discussion of the paper.*

**Results and Discussion : 3.4.1 Representation of natural variability in environmental conditions within the Andean region**

The runoff and erosion processes decrease significatively (excepted exceptional events) with appropriate technique to protect the soil (cultural, terraces, etc.), with vegetal cover (natural or anthropized origin). This is not a surprise, and your review shows that what has already been studied on these "parameters". However, the rainfall characteristics (intensity, concentration of precipitation, etc.) seem little mentioned. Is it correct?

*This is correct : there are few studies that have quantified the effect of rainfall intensity and duration on erosion rates. The few studies that analyzed this effect are based on plot measurements, and (often) simulated rainfall. We will mention this knowledge gap in the conclusion of the study.*

**3.4.2 Gap between plot-scale and catchment-scale erosion assessments**, apparently, we have a "confusion" about erosion processes depending on the studied scale. Micro-plots (≤ 1 m$^{-2}$ to 10 m$^{-2}$), generally the authors are studying Detachability and/or Erodibility expressed in gramme per m$^{-2.}$ Catchment scale the authors are interested by erosion expressed in tons per hectare. Due to this difference in scale, it is therefore very difficult to compare the studies with each other. Can you be more specific on how you compare these two spatial scales?

*In soil erosion studies, different measures of soil erosion are reported: mass loss per surface area per time (M/L²/T), surface lowering per unit of time (L/T), or mass loss measured during rainfall simulations (M/L²). In order to have comparable datasets, we have converted all data to obtain erosion measures in t/km²/yr. We will provide more details on the conversions in the revised document.*

*Although there is a wealth of information on sediment production from plot studies on the one hand, and sediment yield from larger catchments on the other hand, it is not straightforward to make the link between the erosion processes in the sediment sources and the sediment throughput of larger catchments. We agree with the point made by the referee "it is difficult to compare the studies". This is also the point that we want to make in the discussion, i.e. that there is a gap in knowledge on the sediment cascade: sediment transfer, (temporary) storage and deposition.*

**Conclusion and perspectives.**

It is a long conclusion about the contains of this review, maybe too long. On the other hand, the research perspectives look few and insufficiently precise. Could you add/suggest some specific studies should be done on the rainfall intensity thresholds corresponding to the effect of soil erosion parameter. (Which ones? How?)? Is it desirable to mention more studies related to climate change?

*We will revise the conclusion, and include an outlook for future research. Climate variability and climate change is expected to have an impact on rainfall erosivity. Also, freshly exposed, deglaciated terrain is particularly prone to soil erosion.*

**Figure 3 :** A small green circle is in Argentina but apparently far away of Andes mountains. Is it really related with your keywords and/or scientific questions?

*We agree with the referee that this study comes from an area that is far from the other study sites (and might represent another environmental context). We have removed this site from the systematic analysis.*

---

## Author Comment (AC2)

**Point by point response to Referee#2 comments**

Dear Referee#2,

We would like to thank you for the thoughtful comments on our manuscript "**The effect of natural infrastructure on water erosion mitigation in the Andes »** (soil-2021-76). We are pleased to read that you appreciated the systematic review, and the fact that we included supplements that will summarize the underlying datasets.

We take your point that it is necessary to clarify the intervention and control types that we analyzed in this paper, and to introduce them adequately in the framework of Nature-based Solutions (NbS). We will further elaborate the discussion on the NbS, and rework the conclusion so that it is in line with the research questions posed in the introduction.

Please find below a point-by-point response to all the concerns raised and how we will address them. Your original comments are copied below, and we have highlighted in blue color how we will revise the manuscript accordingly. We hope you find our response and changes to the manuscript satisfying and we are looking forward to hearing from you.

**Major comments**

In general, the manuscript is very well written and is well-structured. The objective and research questions are well-defined, but could be better integrated into the Conclusions (see below). The methodology is clearly described and I'm happy that the authors included several supplements, which gives confidence in the robustness of the study. The main critique I have is the way the different interventions are presented in the manuscript, which is often not very consistent. In the abstract (lines 20-21) and the Introduction (lines 76-81), the authors refer to three types of natural infrastructure, i.e. protective vegetation, soil and water conservation measures and adaptation measures that regulate flow and transport of water. The latter category seems to be related to reservoirs and other hydraulic structures, which is often referred to as gray infrastructure. It would be odd to include such interventions in this study given the focus on natural infrastructure. However, in the Material and Methods, a different classification is given (lines 154-156), which includes protection of natural vegetation, forestation and soil and water conservation measures. These are subsequently used throughout the rest of the manuscript. So at the end, no gray infrastructure is included in the analysis. Please revise the abstract and Introduction so that the same categories are used throughout the entire manuscript. Also please revise the terminology used for the different interventions, because there are some inconsistencies throughout the manuscript. And keep a clear difference between the control and intervention categories, sometimes all six categories are discussed as if they were they were all natural infrastructure or interventions.

*We understand that the categorization of the interventions might have been confusing, and we will revise the text. In the original concept of the study, hydraulic infrastructure that regulates flow and transport of water was included because they are de facto considered like "natural infrastructure" in some Andean countries such as Peru. This concerns small pounds (e.g. "qochas") or irrigation systems (e.g. "amunas") often inherited from traditional knowledge. We understand your concern of having them categorized as "natural" rather than "gray" infrastructure.*

*As only very few of these studies on hydraulic infrastructure included quantitative measures on soil erosion/sediment yield, it was not relevant to keep this category in the quantitative analyses. Therefore, we focused the analyses on the categories for which we had sufficient datasets, i.e. protective vegetation (PRO and FOR) and soil and water conservation measures (SWC).*

*We will revise the terminology accordingly, and mention only those control and intervention categories for which we had sufficient quantitative data for statistical analyses.*

The objective of the study is to quantify the effectiveness of natural infrastructure and in several instances a reference is made to the control-intervention design of the majority of the studies included in the systematic review. With this in mind, I do not understand why the authors present the results of the differences between two control types in Figure 5 (i.e. cropland vs. rangeland, cropland vs. bare soil and rangeland vs. bare soil). These results are not in line with the objective. For clarity and consistency, I suggest to remove those results or show them in a supplement, and focus the results on the difference between conventional agricultural practices and natural infrastructure.

*For the analyses of the full-text papers, we used a control-intervention design as is commonly done in environmental assessments. In the Andean region, interventions are often realised on degraded agricultural land. This land can be used for crops or pastures; left as bare fallow or abandoned (Zimmerer, 1993; Henry et al., 2013). We hypothesized that the effectiveness of the intervention will depend on the initial state of the land. That is the reason why we kept traditional agriculture (cropland and rangeland) and bare land (bare) as separate categories of "control" in our analyses.*

*We acknowledge that this might have caused confusion, and will revise the text accordingly. In addition, we will reorder the categories in the figures, so that we have the different interventions (PRO, FOR, and SWC) grouped in one part of the graph, and the three different control types (RANGE, CROP, BARE) in the other part. We will also give the average value for all interventions, and all control types. In Figure 5, we will better differentiate intervention from control.*

The authors mainly focused on the 6 indicators as shown in the figures and tables. Apart from that, the authors recorded some other information regarding climate, soil type, land use, among others. It would be interesting to see how these variables affect the results. This means an additional analysis, but could explain some of the uncertainty. Since there are not many studies to perform such an analysis, the authors could aggregate all categories into two categories, i.e. natural infrastructure and control.

*We agree with the referee that it is worthwhile to analyze whether effect sizes (Figure 5) vary with climate, soil type, and land use history. In the original study, this was not done because of the limited number of observations per category that prohibited subdividing the dataset per climate, soil type or land use (history) classes. For some indicators (like Sloss, SOC or BD) where more datasets are available, such an analysis might be possible when we group interventions and control types. We will elaborate this point in the revisions.*

Regarding the discussion of the results. The interventions studied in the current study are, of course, not only being studied in the Andes, but also in other geographical areas. I suppose similar studies have been performed in other regions. It would be interesting to discuss how the effectiveness of these interventions in the Andes compare to similar interventions in other geographical regions.

*We agree, and will include in our discussion studies from other geographical regions with similar environmental conditions.*

Regarding the Conclusions. The authors have defined three research questions in the last paragraph of the Introduction. This is very convenient for the Conclusions, because you can just literally answer these questions here. However, I have the feeling that the Conclusions mainly focus on the second research question. The other two questions are somewhat discussed, but could get some more attention. I suggest to reduce the conclusions regarding the second research question, mainly focusing on the most relevant results, and answer the other research questions, at least more than has been done now, i.e. in lines 390-394 and 412-414.

*Thank you for this suggestion. We will rework the conclusion so that the three research questions are adequately addressed in the text.*

**Specific comments**

Line 17: I suggest to first introduce the term "natural infrastructure" before using them in these research questions.

*Ok, we will do so.*

Line 24: Do these two values (1.3 and 2.8) belong to, respectively, the two categories protective vegetation and soil and water conservation measures? Please clarify in the text.

*We will clarify this.*

Line 25: Again a range of values is provided, please be more precise about where this range is based on.

*We will clarify this.*

Line 73: Add "as" between "such" and "peatlands".

*Ok, we will do so.*

Lines 73-74: According to the website of the IUCN, natural infrastructure or natural water infrastructure is considered to be a Nature-based Solution (NbS). I suggest to introduce NbS here, because it is an emerging topic that most readers are familiar with. Lines 74-76: This is actually how Cohen-Shacham et al. (2016) refers to NbS. See also previous comment.

*We agree with these statements, and will refer to NbS following Cohen-Shacham et al. (2016).*

Line 110: Please specify which search fields were used, e.g. Article title, Abstract, Keywords, etc.

*Ok, we will do so.*

Lines 129-132: The second criteria should explicitly suggest that modelling studies are included or excluded, now it is a bit unclear if modelling studies are considered or not.

*We will clarify this in the text. Modelling studies were only included in the analyses when the models had been fully calibrated and validated using field experiments from a similar geographic and environmental setting.*

Line 137: So from 813 studies the authors went to 190 studies based on the inclusion and exclusion criteria (lines 129-134), but how were the 53 studies excluded. Here it seems that these were excluded based on the same criteria. Please clarify in the text.

*From the 190 studies, 53 were excluded as they studied only landslides or landslide erosion, and did not contain measurements of soil erosion by water or soil quality. We will clarify this in the text.*

Line 145: Why is altitude not numbered?

*We will correct this. We considered it to be part of the coordinates (X,Y,Z) but will make the necessary changes.*

Lines 151-152: The latter two categories overlap, i.e. large catchment (> 1000 km2) and landscape scale, which is not defined by a study area size. Please be more specific about the difference between these two classes.

*We will clarify this in the text. We made the difference between studies that are organized per plot, per drainage basin (small and large catchment), and studies that contain field measurements that are taken over a larger geographical area. The latter can include measurements from different catchments.*

Lines 154-157: These are the three natural infrastructure categories as defined earlier (lines 94-96)? It seems that the previous three categories were defined differently, i.e. from the previous categories PRO and FOR would be included in category 1, SWC in category 2 and category 3 is not included here. Please clarify in the text.

*We agree that this was confusing, and will rework the text accordingly. See our reply to the main comment above.*

Lines 182-183: Did the authors use specific software to extract the data from figures? I suppose that tables were also used.

*To extract data from figures, we used the software "PlotDigitizer". Information from in-text tables and supplementary material was copied and tabulated in spreadsheets.*

Line 217: Please replace "both" with "all three of them", or similar.

*Ok, we will do so.*

Lines 217-220: The most southeast located study area doesn't seem to be located in the Andes. Please clarify why this study is nevertheless included. Or show in the map what area is considered to be the Andes.

*A similar comment was made by Referee#1. We agree that this study comes from an area that is far from the other study sites (and might represent another environmental context). We have removed this site from the systematic analysis.*

Lines 243-244: Here the conventional/traditional agricultural practices are also included under natural infrastructure. To prevent confusion, please rewrite this sentence.

*Correct, we will rephrase*

Lines 243-246: This sentence needs to be revised. I suppose the authors are referring to the results of the Kruskal-Wallis test, as shown in Figure 4, not in Table 1. Please be more specific about which results is discussed and especially refer to the statistical test.

*We're referring to the results of the Kruskal-Wallis test that are shown in Figure 4. In the text, we will better differentiate the results of the Kruskal-Wallis test, and the posthoc comparison test (shown in Table 1).*

Lines 261-262: Where is this results shown? Pastures and native grasslands are included in any of the categories? Please clarify in the text.

*These results are shown in Figure 5 where we report the effect size for restoration and protection of native vegetation (PRO) compare to rangelands/pastures. The category "PRO" includes native grasslands. We acknowledge that this might have been confusing, and will rephrase the sentence accordingly, and refer to Figure 5.*

Lines 277: With natural vegetation the authors mean the protected areas (i.e. PRO)? Please be very consistent with using the names of the categories. To prevent confusion, I suggest to always use the same name and/or use the abbreviation.

*Correct, with areas covered by natural vegetation, we mean the protected areas (PRO). We will revise the text, and systematically use the same name/abbreviation for the categories.*

Lines 279-280: Is there a separate statistical test performed on the differences between the intervention and control categories? I could not find this in the manuscript. Please clarify in the text.

*This information was not included in the original manuscript, as we focused on the comparison between the six categories. We will add the results of the comparison [all intervention/ all control] in the text.*

Lines 304-305: But Figure 5 includes comparisons of 1 and 2 independent case studies. I suggest to indicate in Figure 5 which comparisons are considered and which not. For instance, the comparisons that are considered could be indicated with a darker color and the ones that are neglected with a lighter color (or with some level of transparency).

*Thank you for this suggestion. We will adapt the figure, and show the results that are based on < 3 case studies in lighter color.*

Line 326: Why are two values shown here, to what are these two values referring to. Please clarify in the text.

*These values are the effect sizes of SWC compared to croplands and rangelands under traditional agricultural management. We will clarify this in the text.*

Line 329: Where are SWC compared to grasslands? In Figure 5 the categories are compared with either cropland or rangeland, not with grassland. Please clarify in the text.

*Many thanks for noting this error, as it should read as "rangelands" and not as "grasslands". We will make the necessary correction in this text.*

Lines 300-343: In Figure 5 the comparison is made between the categories and cropland (upper) and rangeland (lower). This subsection is mainly focused on the differences with cropland and rangeland is only mentioned in one paragraph (lines 307-312). I was under the impression that the authors would

compare the intervention categories (PRO, FOR, SWC) with the control categories (CROP, RANGE, BARE). Which is also much better in line with the objective of the study. So why did the authors compare, for instance, rangeland and bare soil with cropland? And why are the intervention categories not compared with bare soil, in a similar way as has been done with cropland and rangeland?

*We take your point, and appreciate this suggestion. In fact, we made the comparison BARE-CROP and BARE-RANGE, as this is often done in soil erosion studies based on e.g. Wishmeier-type experiments (De Noni et al., 2001). The number of case-studies that make this comparison is rather high, which allowed us to do statistics on the effect sizes.*

*However, we agree that it would be more relevant for this study to analyse the effect size of the three intervention types compared to the control types, even if this is based on a lower number of case-studies. We will re-organise the presentation of the results, and make the comparison between the three intervention categories (PRO, FOR, SWC) and the three control categories (CROP, RANGE, BARE). This will result in a new figure with 9 panels (3 by 3).*

Line 369: With "both" the authors mean "simultaneously"?

*Correct, we mean studies that have looked at water erosion at the catchment scale, and that at the same time realised measurement of soil erosion at the plot scale in the same catchment. We will adapt the text to clarify this.*

Line 383: I think that this should be 43%, instead of 40%.

*We wrote "more than 40%", but will replace this in the text by "43%" which is the exact number.*

Lines 392-393: These two values (i.e. 85 and 125) are referring to the previous sentence? Please combine these two sentences into one sentence to know where the values are referring to.

*We take your point, and will combine the information of the two sentences into one sentence.*

Lines 403-407: The authors frequently indicate the effect size results as a range between the minimum and maximum value of the error bars in Figure 5. I think it would increase the readability if the authors would indicate the mean response ratio in the text, rather than the spread of the error. For instance, as has been done in the subsequent two sentences.

*The two values that are reported are the mean effect sizes compared to cropland and rangeland under traditional agricultural management. We acknowledge that this might have been confusing, and will make the necessary corrections. In the revised text, we will better separate the effect size compared to cropland, rangeland and bare soil; and discuss them separately as the effect size depends on the initial state of the land.*

**Figures and Tables**

Figures: Please increase the font size of the smallest font.

*Ok, we will do so.*

Figure 3: I suggest to include the names of the countries in the map, for those readers that are not too familiar with the topography of South America. Please, include a reference to the DEM used as background for the map.

*Ok, we will do so.*

Table 1: Please replace the last sentence with something like this: "The box-plots followed by a common letter are not significantly different by the Dunn's posthoc test at the 0.05 level of significance." See Piepho (2018) for an interesting discussion about the meaning of these letters and for suggestions on how to refer to them in table and figure captions.

*Thank you for this suggestion, and for the reference to this paper. We will make the necessary changes to the text.*

**Supplement**

Supplement C: I highly encourage to include the underlying data as a supplement to this study. However, it seems that some information is missing in this spread sheet. It would be useful to include a separate sheet where the different codes (e.g. for the ecosystem services, natural infrastructure and treatment) are explained. Also the actual data is missing, there are no values included on the right side of 5. Indicators. Please explain why these have not been included.

*We will provide the supplementary material with the underlying data when the paper is accepted for publication.*

**References**

Piepho, H.-P.: Letters in Mean Comparisons: What They Do and Don't Mean, Agron. J., 110(2), 431–434, doi:10.2134/agronj2017.10.0580, 2018.

*Thanks for this suggestion. We will check the document.*

*References*

*Cohen-Shacham, E., Walters, G., Janzen, C., and Maginnis, S.: Nature-based Solutions to address global societal challenges, Gland, Switzerland: IUCN, 97pp., 2016*

*De Noni, G., Viennot, M., Asseline, J., and Trujillo, G.: Terre d'altitude, terres de risque. La lutte contre l'érosion dans les Andes équatoriennes, IRD éditions, Collection Latitudes 23, Paris, France, 2001.*

*Henry, A., Mabit, L., Jaramillo, R. E., Cartagena, Y., and Lynch, J. P.: Land Use Effects on Erosion and Carbon Storage of the Río Chimbo Watershed, Ecuador, Plant and Soil., 367(1–2), 477–491, https://doi.org/10.1007/s11104-012-1478-y, 2013.*

*Zimmerer, K. S.: Soil Erosion and Labor Shortages in the Andes with Special Reference to Bolivia, 1953-1991: Implications for 'Conservation-with-Development', World Development., 21, 1659–1675, https://doi.org/10.1016/0305-750X(93)90100-N, 1993.*

---

## Author Response (AR1)

Dear Referees,

We would like to thank you for the thoughtful comments on our manuscript "**The effect of natural infrastructure on water erosion mitigation in the Andes »** (soil-2021-76). Please find below a point-by-point response to all the concerns raised and how we addressed them. Your original comments are copied below, and we have highlighted in blue color how we revised the manuscript accordingly. We hope you find our response and changes to the manuscript satisfying and we are looking forward to hearing from you.

With kind regards,
Veerle Vanacker on behalf of all co-authors

**Point by point response to Referee#1 comments**

**Major comments**
- Weak point: some information or references about sites/countries and rain characteristics are unclear or missing because the research of the studies is done by a specific selection of key words.
- Strong point: the overview of the references is well analysed.

*We take your point on the necessity to include more information about the overall characteristics of the study sites, like rainfall amounts or intensities. In a new figure, Figure 6, we now show the distribution of the records (and the Andean region) with regard to precipitation amount, altitude, and slope gradient; and discuss the representativeness of the studies on L376-383.*

With some improvements, especially verifying the interest of specific references, in the definition of the erosion/erodibility parameters, and the conclusion-perspectives this paper could provide a relevant review for the research community working on different aspect of Andes mountains "ecology" (interaction soil, water, vegetation) and more generally for erosion processes studies.

*Thank you for the suggestions. In the revised manuscript, we included the references on soil erosion and erodibility in the Andes. We also expanded the discussion, providing guidelines for future erosion studies in L405-414.*

**Specific comments**
Line 54: a large research and applied project was conducted in Ecuador to protect Quito City, cf. *Perrin JL, Bouvier C, Janeau JL, Menez G, Cruz F. Rainfall/runoff processes in a small peri-urban catchment in the Andes mountains. The Rumihurcu Quebrada, Quito (Ecuador). Hydrological processes 2001; 15: 843-854.* I noted that you mentioned this study in the supplement D.

*We thank the referee for this suggestion. The study is included in the compilation, and we refer to this study in L298.*

Line 102 then line 110 to 115 and Supplement C:

I can understand that you selected studies due to absence of quantitative on-site or off-site soil erosion or soil quality measurements and I know that it is difficult to collect all the studies carried out on your topic, however it seems that some studies, are missing despite using your key words:

Buytaert W, Wyseure G, De Bievre B, Deckers J. The effect of **land-use** changes on the hydrological behaviour of Histic Andosols in south Ecuador. Hydrological processes 2005; 19: 3985-3997.

Janeau JL, Grellier S, Podwojewski P. Influence of rainfall interception by endemic plants versus short cycle crops on water **infiltration** in high altitude ecosystems of Ecuador. Hydrology Research 2015; 46: 1008-1018.

Poulenard J, Podwojewski P, Janeau JL, Collinet J. Runoff and **soil erosion** under rainfall simulation of Andisols from the Ecuadorian Paramo: effect of tillage and burning. Catena 2001; 45: 185-207.

*We are aware of these studies, as they were part of the peer-reviewed studies that we initially retrieved from SCOPUS. The three studies are highly relevant for studying rainfall-runoff processes in the Andes. In the screening phase, we removed studies that did not contain quantitative data on soil erosion (soil erosion rates measured as M/L²/T or L/T). Thereby, the database was reduced to 118 studies.*

*We included reference to the work of Poulenard et al. (2001) and Janeau et al. (2015) in the text (L331, 332, 407) as these studies include empirical data on rainfall-runoff behavior in paramo ecosystems. We did not reference all 118 papers in the text, but list them in Supplement D.*

**Data base development – line 161 to 170:** In this paragraph, I appreciated the definition of the soil indicators, but I suggest providing more explanation for the role of the rain (different intensities and duration depending on altitude…) and the role of the vegetal cover (throughfall/stemflow for infiltration, soil protection by different type of covers…), the broad range of bulk density (type of volcanic ashes). This complement of explanation could be useful to explain your result Line 284 to 291 in Results and Discussion and your Figure 4

*We thank the referee for this suggestion. Part of the variability in response ratios (shown in Figure 5) can be attributed to these factors. Given the relatively small dataset, it is not possible to analyse quantitatively the role of rain, vegetation cover or parent material. In the revised version of the text, we expanded the discussion based on existing literature from case-studies. For example, on L330-335, we mention the role of native vegetation in facilitating soil water infiltration through stemflow.*

**Results and Discussion : 3.4.1 Representation of natural variability in environmental conditions within the Andean region**

The runoff and erosion processes decrease significatively (excepted exceptional events) with appropriate technique to protect the soil (cultural, terraces, etc.), with vegetal cover (natural or anthropized origin). This is not a surprise, and your review shows that what has already been studied on these "parameters". However, the rainfall characteristics (intensity, concentration of precipitation, etc.) seem little mentioned. Is it correct?

*This is correct : there are few studies that have quantified the effect of rainfall intensity and duration on erosion rates. The few studies that analyzed this effect are based on plot measurements, and (often) simulated rainfall. We have now mentioned this knowledge gap in the conclusion of the study (L452-456).*

**3.4.2 Gap between plot-scale and catchment-scale erosion assessments**, apparently, we have a "confusion" about erosion processes depending on the studied scale. Micro-plots ($\leq 1\ m^{-2}$ to $10\ m^{-2}$), generally the authors are studying Detachability and/or Erodibility expressed in gramme per $m^{-2.}$ Catchment scale the authors are interested by erosion expressed in tons per hectare. Due to this

difference in scale, it is therefore very difficult to compare the studies with each other. Can you be more specific on how you compare these two spatial scales?

*In soil erosion studies, different measures of soil erosion are reported: mass loss per surface area per time (M/L²/T), surface lowering per unit of time (L/T), or mass loss measured during rainfall simulations (M/L²). In order to have comparable datasets, we have converted all data to obtain erosion measures in t/km²/yr.*

*Although there is a wealth of information on sediment production from plot studies and sediment yield from larger catchments, it is not straightforward to make the link between the erosion processes in the sediment sources and the sediment throughput of larger catchments. We agree with the point made by the referee "it is difficult to compare the studies". This is also the point that we made in the discussion, i.e. that there is a gap in knowledge on the sediment cascade: sediment transfer, (temporary) storage and deposition. We have further elaborated this part in section 3.4.2 (L405-407; and L409-413).*

**Conclusion and perspectives.**

It is a long conclusion about the contains of this review, maybe too long. On the other hand, the research perspectives look few and insufficiently precise. Could you add/suggest some specific studies should be done on the rainfall intensity thresholds corresponding to the effect of soil erosion parameter. (Which ones? How?)? Is it desirable to mention more studies related to climate change?

*We carefully revised the conclusion, and included a paragraph on future outlooks (L449-454). Climate variability and climate change is expected to have an impact on rainfall erosivity. Also, freshly exposed, deglaciated terrain is particularly prone to soil erosion.*

**Figure 3 :** A small green circle is in Argentina but apparently far away of Andes mountains. Is it really related with your keywords and/or scientific questions?

*We agree with the referee that this study comes from an area that is far from the other study sites (and might represent another environmental context). We have removed this site (Bujan et al., 2000) from the systematic analysis, by excluding it in the eligibility stage (full text papers excluded : 54 instead of 53). The final number of studies in the quantitative analyses was 118 (instead of 121), as we noticed that there were two records for which we didn't have quantitative data on soil erosion rates in $M.L^{-2}.T^{-1}$ or $L.T^{-1}$. The number of records in Figure 1 (flowchart) was modified, as well as the map and circle diagram in Figure 3.*

*Accordingly, the analyses were done on the reduced database. All tables and figures have been modified, including Table 3, Figure 4 and Figure 5. The necessary modifications were done in the text.*

*Bujan, A., Santanatoglia, O J., Chagas, C., Massobrio, M., Castiglioni, M. (2000). Preliminary study on the use of the 137Cs method for soil erosion investigation in the pampean region of Argentia. Acta Geologica Hispanica 35(3-4), 271-277.*

**Point by point response to Referee#2 comments**

**Major comments**

In general, the manuscript is very well written and is well-structured. The objective and research questions are well-defined, but could be better integrated into the Conclusions (see below). The methodology is clearly described and I'm happy that the authors included several supplements, which gives confidence in the robustness of the study. The main critique I have is the way the different interventions are presented in the manuscript, which is often not very consistent. In the abstract (lines 20-21) and the Introduction (lines 76-81), the authors refer to three types of natural infrastructure, i.e. protective vegetation, soil and water conservation measures and adaptation measures that regulate flow and transport of water. The latter category seems to be related to reservoirs and other hydraulic structures, which is often referred to as gray infrastructure. It would be odd to include such interventions in this study given the focus on natural infrastructure. However, in the Material and Methods, a different classification is given (lines 154-156), which includes protection of natural vegetation, forestation and soil and water conservation measures. These are subsequently used throughout the rest of the manuscript. So at the end, no gray infrastructure is included in the analysis. Please revise the abstract and Introduction so that the same categories are used throughout the entire manuscript. Also please revise the terminology used for the different interventions, because there are some inconsistencies throughout the manuscript. And keep a clear difference between the control and intervention categories, sometimes all six categories are discussed as if they were they were all natural infrastructure or interventions.

*We understand that the categorization of the interventions might have been confusing, and we revised the text, tables and figures. In the original concept of the study, hydraulic infrastructure that regulates flow and transport of water was included because they are de facto considered like "natural infrastructure" in some Andean countries such as Peru. This concerns small pounds (e.g. "qochas") or irrigation systems (e.g. "amunas") often inherited from traditional knowledge. We understand your concern of having them categorized as "natural" rather than "gray" infrastructure. As only very few of these studies on hydraulic infrastructure included quantitative measures on soil erosion/sediment yield, it was not relevant to keep this category in the quantitative analyses. We focused the analyses on the categories for which we had sufficient datasets, i.e. protective vegetation (PRO and FOR) and soil and water conservation measures (SWC). These three types of interventions are the ones that are now fully discussed in the revised version of the paper.*

*Per your suggestion, we revised the text of the abstract and introduction. We now refer to the three categories of intervention for which we had sufficient data: PRO, FOR and SWC.*

*Changes were made in the abstract (L24-26): "Three major categories of natural infrastructure were considered: restoration and protection of natural vegetation such as forest or native grasslands, forestation with native or exotic species, and implementation of soil and water conservation measures for erosion mitigation."*

*Also in the introduction (L83-85): "In the Andean context, three large groups of water-related interventions can be identified: interventions based on land use and protective land cover including (1) restoration and protection of native ecosystems such as montane forests or grasslands and (2) forestation with native or exotic species, and (3) soil and water conservation measures including crop*

*management, conservation tillage, and slow forming terraces, and the implementation of linear elements such as vegetation strips and check dams."*

*And materials and methods (L102-104): "We adapted the typology to the Andean region and defined three large groups of interventions: (i) the restoration and protection of native ecosystems, (ii) the forestation with native or exotic species and (iii) the implementation of soil and water conservation measures."*

*Accordingly, we revised Table 1 and re-organised the table per intervention and control type. The same thing was done for Figure 4, where intervention and control types were grouped, so that the information of the treatment and control groups are better separated from each other.*

*In Figure 5 with the response ratio, we focused on the comparisons between the three types of intervention and one control group (including cropland, grassland and bare land). We excluded the comparison between the three control types, as this was not relevant for the research questions posed in this paper.*

The objective of the study is to quantify the effectiveness of natural infrastructure and in several instances a reference is made to the control-intervention design of the majority of the studies included in the systematic review. With this in mind, I do not understand why the authors present the results of the differences between two control types in Figure 5 (i.e. cropland vs. rangeland, cropland vs. bare soil and rangeland vs. bare soil). These results are not in line with the objective. For clarity and consistency, I suggest to remove those results or show them in a supplement, and focus the results on the difference between conventional agricultural practices and natural infrastructure.

*For the analyses of the full-text papers, we used a control-intervention design as is commonly done in environmental assessments. In the Andean region, interventions are often realised on degraded agricultural land. This land can be used for crops or pastures; left as bare fallow or abandoned (Zimmerer, 1993; Henry et al., 2013). We hypothesized that the effectiveness of the intervention will depend on the initial state of the land. That is the reason why we kept traditional agriculture (cropland and rangeland) and bare land (bare) as separate categories of "control" in our analyses.*

*We acknowledge that this might have caused confusion. As the number of observations per control type is very low, we followed your advice and combined all control types (RANGE, CROP, BARE) in one "control group". We have re-run the analyses (see L211-213 in the method). The new version of Figure 5 shows the effect size of the three interventions compared to the control group, and Table 2 resumes the statistics of the response ratios per indicator.*

*The text of the discussion and conclusion was thoroughly revised, to be consistent with the three treatment-control comparisons (L309-316, Figure 5, Table 2, L317-335, L338-340, L350-362)*

The authors mainly focused on the 6 indicators as shown in the figures and tables. Apart from that, the authors recorded some other information regarding climate, soil type, land use, among others. It would be interesting to see how these variables affect the results. This means an additional analysis, but could explain some of the uncertainty. Since there are not many studies to perform such an analysis, the authors could aggregate all categories into two categories, i.e. natural infrastructure and control.

*We followed the referee's advice and regrouped all control types into one "control group". Table 2 resumes the statistics of the treatment-control comparisons. Given the low number of observations for most indicators, the scatter in the response ratios is important for e.g. RC or Sloss. In addition, the records are not well distributed in the geographical space, and there exists spatial bias. We have elaborated this point in the section 3.4.1 (L373-384).*

*For these reasons, we think that it is not pertinent to draw conclusions on the role of climate, soil type or land use based on this compilation. More empirical work is necessary to cover the full range of climate, land use and topographic settings.*

Regarding the discussion of the results. The interventions studied in the current study are, of course, not only being studied in the Andes, but also in other geographical areas. I suppose similar studies have been performed in other regions. It would be interesting to discuss how the effectiveness of these interventions in the Andes compare to similar interventions in other geographical regions.

*We have included additional references, mainly from the Andean region. It is not evident to compare natural infrastructure interventions between geographical regions, as the specific type of interventions is often region-specific (e.g. qochas, amunas).*

Regarding the Conclusions. The authors have defined three research questions in the last paragraph of the Introduction. This is very convenient for the Conclusions, because you can just literally answer these questions here. However, I have the feeling that the Conclusions mainly focus on the second research question. The other two questions are somewhat discussed, but could get some more attention. I suggest to reduce the conclusions regarding the second research question, mainly focusing on the most relevant results, and answer the other research questions, at least more than has been done now, i.e. in lines 390-394 and 412-414.

*Thank you for this suggestion. We have reworked the conclusions, and have now drafted three paragraphs that respond to the research questions posed in the introduction (see L433-446).*

**Specific comments**

Line 17: I suggest to first introduce the term "natural infrastructure" before using them in these research questions.

*We added two sentences that introduced the concept of "natural infrastructure" (L78-81).*

*« There exists a renewed interest in catchment management based on natural infrastructure. Natural infrastructure is defined by the IUCN as "services that nature provides such as peatlands sequestering carbon, lakes storing large water supplies, and floodplains absorbing excess water runoff".*

Line 24: Do these two values (1.3 and 2.8) belong to, respectively, the two categories protective vegetation and soil and water conservation measures? Please clarify in the text.

*As we combined all control types into one control group, this is not relevant anymore. We made the necessary corrections to the text.*

Line 25: Again a range of values is provided, please be more precise about where this range is based on.

*We made the necessary corrections to the text.*

Line 73: Add "as" between "such" and "peatlands".

*Done.*

Lines 73-74: According to the website of the IUCN, natural infrastructure or natural water infrastructure is considered to be a Nature-based Solution (NbS). I suggest to introduce NbS here, because it is an emerging topic that most readers are familiar with.

*Done.*

Lines 74-76: This is actually how Cohen-Shacham et al. (2016) refers to NbS. See also previous comment.

*We rephrased this part to clarify the link between natural infrastructure and nature-based solutions. (L78-82)*

*"Soil conservation measures are receiving renewed interest in the context of nature-based solutions. They are defined by the IUCN as "services that nature provides such as peatlands sequestering carbon, lakes storing large water supplies, and floodplains absorbing excess water runoff" (Cohen-Shacham et al., 2016). Natural infrastructure is part of nature-based solutions, and their infrastructure-like function helps to protect, sustainably manage or restore ecosystems while simultaneously providing human well-being and biodiversity benefits".*

Line 110: Please specify which search fields were used, e.g. Article title, Abstract, Keywords, etc.

*We specified that we searched in the title, abstract and keywords: (L116)*

*"We searched within the article title, abstract and keywords for the following terms".*

Lines 129-132: The second criteria should explicitly suggest that modelling studies are included or excluded, now it is a bit unclear if modelling studies are considered or not.

*We clarified this in the text (L137-138). Modelling studies were only included in the analyses when the models had been fully calibrated and validated using field experiments from a similar geographic and environmental setting.*

*"they are experimental studies including observational datasets or are modelling studies that are fully validated with field experiments or measurements".*

Line 137: So from 813 studies the authors went to 190 studies based on the inclusion and exclusion criteria (lines 129-134), but how were the 53 studies excluded. Here it seems that these were excluded based on the same criteria. Please clarify in the text.

*From the 190 studies that were assessed in full-text, we excluded 54 studies as they did not contain quantitative measures of erosion rates or soil quality for different intervention and control types. This mainly concerned descriptive studies on landslides or landslide erosion. See L144-147.*

*« We assessed 190 studies in full-text, and further excluded 54 papers as the studies did not report quantitative measures of erosion rates or soil quality for different classes of land use and protective land cover, soil and water conservation measures, or elements of hydraulic regulation. At this stage, this mainly concerned scientific reports on landslides and landslide-related erosion events »*

Line 145: Why is altitude not numbered?

*We solved this in L152 : "We recorded the following ancillary geographic data: (1) country, (2) site name, (3) coordinates (latitude and longitude in decimal degrees), (4) altitude (meters above sea level, m a.s.l.), and information on (4) bioclimate, (5) surface lithology, (6) ecosystem and (7) landform".*

Lines 151-152: The latter two categories overlap, i.e. large catchment (> 1000 km2) and landscape scale, which is not defined by a study area size. Please be more specific about the difference between these two classes.

*We clarified this in the text L158-160. We made the difference between studies that are organized per plot, per drainage basin (small and large catchment), and studies that contain field measurements that are taken over a larger geographical area:*

*"The latter contain data collections that are not organised by hydrological units, and that include measurements taken over a larger geographical area. »*

Lines 154-157: These are the three natural infrastructure categories as defined earlier (lines 94-96)? It seems that the previous three categories were defined differently, i.e. from the previous categories PRO and FOR would be included in category 1, SWC in category 2 and category 3 is not included here. Please clarify in the text.

*We reworked the text, and made changes to the introduction to avoid confusion. See our reply to the main comment above.*

Lines 182-183: Did the authors use specific software to extract the data from figures? I suppose that tables were also used.

*We clarified this in the text "Mean, sample size and deviation metrics were extracted from figures using PlotDigitizer. Information from in-text tables and supplementary material was copied and tabulated in spreadsheets". L190-191*

Line 217: Please replace "both" with "all three of them", or similar.

*Thanks, we rephrased L227 : "all three of them (SWC, PRO and FOR).*

Lines 217-220: The most southeast located study area doesn't seem to be located in the Andes. Please clarify why this study is nevertheless included. Or show in the map what area is considered to be the Andes.

*A similar comment was made by Referee#1. We agree that this study comes from an area that is far from the other study sites (and might represent another environmental context). We have removed this site from the systematic analysis, the analyses were re-run, figures and tables and text was adapted.*

Lines 243-244: Here the conventional/traditional agricultural practices are also included under natural infrastructure. To prevent confusion, please rewrite this sentence.

*We have reorganized the analyses, and this is now solved.*

Lines 243-246: This sentence needs to be revised. I suppose the authors are referring to the results of the Kruskal-Wallis test, as shown in Figure 4, not in Table 1. Please be more specific about which results is discussed and especially refer to the statistical test.

*We were referring to the results of the Kruskal-Wallis test that are shown in Figure 4. As the database was adapted, we have re-run the analyses using an updated version of the PMCMR package, "PMCMRplus". We used the Kruskal-Wallis rank sum test, and the Dunn's non-parametric all-pairs comparison test at 5% level of significance. We read the paper of Piepho (2018) and are now more specific about the tests that were used (L242-244).*

Lines 261-262: Where is this results shown? Pastures and native grasslands are included in any of the categories? Please clarify in the text.

*These results are shown in Figure 5 where we report the effect size for restoration and protection of native vegetation (PRO) compared to the control group. Note that we now aggregated all control types into one control group (cropland+grassland+bare land). The category "PRO" includes native grasslands. We have rephrased the text to avoid confusion between native grassland and pastures.*

Lines 277: With natural vegetation the authors mean the protected areas (i.e. PRO)? Please be very consistent with using the names of the categories. To prevent confusion, I suggest to always use the same name and/or use the abbreviation.

*Correct, with areas covered by natural vegetation, we mean the protected areas (PRO). We revised the text carefully to avoid confusion.*

Lines 279-280: Is there a separate statistical test performed on the differences between the intervention and control categories? I could not find this in the manuscript. Please clarify in the text.

*Given the limited number of records in the compilation, we focused now on treatment-control comparisons. We rephrased the text accordingly. In the discussion (Section 3.3) we mention which intervention types has the strongest effect on soil quality and erosion mitigation (see L324-326, and L336-340).*

Lines 304-305: But Figure 5 includes comparisons of 1 and 2 independent case studies. I suggest to indicate in Figure 5 which comparisons are considered and which not. For instance, the comparisons that are considered could be indicated with a darker color and the ones that are neglected with a lighter color (or with some level of transparency).

*Thank you for this suggestion. We adapted the figure, and now show the comparison that are based on less than 4 records in lighter color.*

Line 326: Why are two values shown here, to what are these two values referring to. Please clarify in the text.

*As we regrouped the control types, this is not relevant anymore*

Line 329: Where are SWC compared to grasslands? In Figure 5 the categories are compared with either cropland or rangeland, not with grassland. Please clarify in the text.

*Many thanks for noting this error, as it should read as "rangelands" and not as "grasslands". We corrected this in the text.*

Lines 300-343: In Figure 5 the comparison is made between the categories and cropland (upper) and rangeland (lower). This subsection is mainly focused on the differences with cropland and rangeland is only mentioned in one paragraph (lines 307-312). I was under the impression that the authors would

compare the intervention categories (PRO, FOR, SWC) with the control categories (CROP, RANGE, BARE). Which is also much better in line with the objective of the study. So why did the authors compare, for instance, rangeland and bare soil with cropland? And why are the intervention categories not compared with bare soil, in a similar way as has been done with cropland and rangeland?

*We take your point, and appreciate this suggestion. In fact, we made the comparison BARE-CROP and BARE-RANGE, as this is often done in soil erosion studies based on e.g. Wishmeier-type experiments (De Noni et al., 2001). The number of case-studies that make this comparison is rather high, which allowed us to do statistics on the effect sizes.*

*We agree that it is more relevant to compare treatments with one control group. We reorganized the database, and have re-run the analyses. As mentioned before, new figures and tables were prepared, and the text was adapted accordingly. Figure 5 now shows three panels with the effect size of the three intervention types.*

Line 369: With "both" the authors mean "simultaneously"?

*Correct, we mean studies that have looked at water erosion at the catchment scale, and that at the same time realised measurement of soil erosion at the plot scale in the same catchment. We corrected this. (L411)*

Line 383: I think that this should be 43%, instead of 40%.

*The text was rewritten.*

Lines 392-393: These two values (i.e. 85 and 125) are referring to the previous sentence? Please combine these two sentences into one sentence to know where the values are referring to.

*The text was rewritten.*

Lines 403-407: The authors frequently indicate the effect size results as a range between the minimum and maximum value of the error bars in Figure 5. I think it would increase the readability if the authors would indicate the mean response ratio in the text, rather than the spread of the error. For instance, as has been done in the subsequent two sentences.

*The two values that were reported were the mean effect sizes compared to cropland and rangeland under traditional agricultural management. As we regrouped the control types in one control group, this is not relevant anymore. We now report effect sizes of the three interventions compared to one control group.*

**Figures and Tables**

Figures: Please increase the font size of the smallest font.

*Done*

Figure 3: I suggest to include the names of the countries in the map, for those readers that are not too familiar with the topography of South America. Please, include a reference to the DEM used as background for the map.

*Done*

Table 1: Please replace the last sentence with something like this: "The box-plots followed by a common letter are not significantly different by the Dunn's posthoc test at the 0.05 level of significance." See Piepho (2018) for an interesting discussion about the meaning of these letters and for suggestions on how to refer to them in table and figure captions.

*Thank you for this suggestion, and for the reference to this paper. We made the necessary changes to the text (Caption Table 1; L266, L272, L290).*

**Supplement**

Supplement C: I highly encourage to include the underlying data as a supplement to this study. However, it seems that some information is missing in this spread sheet. It would be useful to include a separate sheet where the different codes (e.g. for the ecosystem services, natural infrastructure and treatment) are explained. Also the actual data is missing, there are no values included on the right side of 5. Indicators. Please explain why these have not been included.

*We will provide the supplementary material with the underlying data when the paper is accepted for publication.*

**References**

Piepho, H.-P.: Letters in Mean Comparisons: What They Do and Don't Mean, Agron. J., 110(2), 431–434, doi:10.2134/agronj2017.10.0580, 2018.

*Thanks for this suggestion. We have used this reference to improve the way we report the results of the statistical tests.*

***References***

*Cohen-Shacham, E., Walters, G., Janzen, C., and Maginnis, S.: Nature-based Solutions to address global societal challenges, Gland, Switzerland: IUCN, 97pp., 2016*

*De Noni, G., Viennot, M., Asseline, J., and Trujillo, G.: Terre d'altitude, terres de risque. La lutte contre l'érosion dans les Andes équatoriennes, IRD éditions, Collection Latitudes 23, Paris, France, 2001.*

*Henry, A., Mabit, L., Jaramillo, R. E., Cartagena, Y., and Lynch, J. P.: Land Use Effects on Erosion and Carbon Storage of the Río Chimbo Watershed, Ecuador, Plant and Soil., 367(1–2), 477–491, https://doi.org/10.1007/s11104-012-1478-y, 2013.*

*Zimmerer, K. S.: Soil Erosion and Labor Shortages in the Andes with Special Reference to Bolivia, 1953-1991: Implications for 'Conservation-with-Development', World Development., 21, 1659–1675, https://doi.org/10.1016/0305-750X(93)90100-N, 1993.*